



# Inter-comparison of snow depth retrievals over Arctic sea ice from radar data acquired by Operation IceBridge

Ron Kwok[1]*, Nathan T. Kurtz[2], Ludovic Brucker[2], Alvaro Ivanoff[3], Thomas Newman[4], Sinead L. Farrell[5,6], Joshua King[7], Stephen Howell[7], Melinda A.Webster[2], John Paden[8], Carl Leuschen[8], Joseph A. MacGregor[2], Jacqueline Richter-Menge[9], Jeremy Harbeck[3], Mark Tschudi[10]

[1]Jet Propulsion Laboratory, California Institute of Technology, Pasadena, California, USA
[2]Cryospheric Sciences Laboratory, NASA Goddard Space Flight Center, Greenbelt, Maryland, USA
[3]ADNET Systems Inc., Lanham, MD, USA
[4]University of Toronto, Toronto, Ontario, Canada
[5]Earth System Science Interdisciplinary Center, University of Maryland, College Park, Maryland, USA
[6]Laboratory for Satellite Altimetry, Satellite Oceanography and Climatology Division, NOAA Center for Weather and Climate Prediction, College Park, Maryland, USA
[7]Climate Research Division, Environment and Climate Change Canada, Toronto, Ontario, Canada
[8]Center for Remote Sensing of Ice Sheets, The University of Kansas, Lawrence, Kansas, USA
[9]University of Alaska Fairbanks, Fairbanks, Alaska, USA
[10]University of Colorado, Boulder, Colorado, USA

*Correspondence to*: Ron Kwok (ron.kwok@jpl.nasa.gov)

**Abstract.** Since 2009, the ultra-wideband snow-radar on Operation IceBridge has acquired data in annual campaigns conducted during the Arctic and Antarctic springs. Progressive improvements in radar hardware and data processing methodologies have led to improved data quality for subsequent retrieval of snow depth. Existing retrieval algorithms differ in the way the air-snow and snow-ice interfaces are detected and localized in the radar returns, and in how the system limitations are addressed (e.g., noise, resolution). In 2014, the Snow Thickness On Sea Ice Working Group (STOSIWG) was formed and tasked with investigating how radar data quality affect snow depth retrievals and how retrievals from the various algorithms differ. The goal is to understand the limitations of the estimates and to produce a well-documented, long-term record that can be used for understanding broader changes in the Arctic climate system. Here, we assess five retrieval algorithms by comparisons with field measurements from two ground-based campaigns, including the BRomine Ozone Mercury EXperiment (BROMEX) at Barrow, Alaska and a field program by Environment and Climate Change Canada (ECCC) at Eureka, Nunavut, available climatology and snowfall from ERA-Interim reanalysis. The aim is to examine available algorithms and to use the assessment results to inform the development of future approaches. We present results from these assessments and highlight key considerations for the production of a long-term, calibrated geophysical record of springtime snow thickness over Arctic sea ice.




## 1 Introduction

The snow layer atop Arctic sea ice modulates the thickness of the underlying ice cover (Maykut and Untersteiner, 1971). In winter, the insulating effects of snow regulate the surface heat balance and hence the rate of ice growth. In spring, the presence of snow shields the ice surface from solar radiation, delaying the onset of surface ice melt. During the melt season, water from snowmelt pools into depressions to form melt ponds, which lower the surface albedo and further enhance the sea ice melt rate. This creates a strong positive feedback between the absorbed downwelling shortwave radiation and melt pond coverage on the ice surface (Kwok and Untersteiner, 2011). Further, available meltwater spreads over larger areas on smoother, seasonal ice compared to rougher, deformed ice (e.g., Fetterer and Untersteiner, 1998; Polashenski et al., 2012; Webster et al., 2015). When this meltwater drains through the ice cover and into the surface ocean (Polashenski et al., 2012), it decreases the salinity and density structure of the ocean, thereby affecting stratification and mixing.

From a remote sensing perspective, estimates of snow loading are required for accurate retrievals of ice thickness from sea ice freeboard (Kwok, 2010). Current and planned satellite altimeters provide only sea ice freeboard (e.g., radar altimeters on CryoSat-2, AltiKa, Sentinel-3) or the combined snow and ice freeboard (lidars on ICESat, ICESat-2), with snow loading left to be measured or modeled by other methods. However, routine measurements of snow depth and density over the Arctic Ocean are not available. Hence, there is extensive interest in the climatology, seasonal and inter-annual variability, and spatial distribution of snow depth for forecast of sea ice behavior, ice thickness retrievals, and for climate analyses and modeling.

Previous understanding of snow depth over Arctic sea ice has been derived from various field surveys (e.g., Sturm et al., 2002b; Sturm et al., 2006) and the climatology based on snow data from drifting stations that operated in 1937 and 1954-1991 (Warren et al. (1999), henceforth, *W99*). Because of the wide-ranging importance of snow depth, remote determination of snow depth at almost any spatial scale has long been desired. The sensor suite of Operation IceBridge (OIB) (Koenig et al., 2010) includes an ultra-wideband snow radar that allows estimates of snow depth by resolving the range location of the air–snow and snow–ice interfaces. Early examination shows that snow depth can be estimated to an uncertainty of about several centimeters, and that the mean snow depth is broadly consistent with the W99 climatology except over seasonal sea ice (e.g., Kurtz and Farrell, 2011; Kwok et al., 2011; Farrell et al., 2012). To date, OIB has acquired eight years (2009–2017) of radar data, including repeat surveys of the early spring snow and ice conditions in different parts of the Arctic. These snow radar datasets provide an unparalleled opportunity to examine snow depth across the Arctic Ocean and both its recent spatial and interannual variability.

Multiple algorithms now exist for the retrieval of snow depth, but they differ in how the air-snow and snow-ice interfaces are detected and localized in the radar returns, and in how the system limitations are accounted for (e.g., noise, resolution). In 2014, a working group (STOSIWG) was formed and tasked with investigating how radar data quality affect snow depth retrievals and how retrievals from the various algorithms differ. In this paper, we report on the assessment of retrievals from five algorithms by comparisons of retrieved snow depths with each other, with measurements from two field surveys, with modified climatology, and with snowfall from ERA-Interim products. The comparisons with field





measurements allow a detailed assessment of the retrievals locally, while the comparisons with climatology and analyzed snowfall provide a large-scale multi-year perspective of their year-to-year retrieval consistency and robustness to changes in radar parameters, and their relative agreements with basin-scale fields. The aim of this paper is to examine these algorithms and to use the assessment results to inform the development of the next generation algorithm. The paper is organized as

follows. The next section summarizes the snow radar, the five retrieval algorithms, and the data sets against which we compare the snow radar data. The data sets include snow measurements from the two field campaigns, modified snow climatology, and snow depth estimates based on snowfall from ERA-Interim products. The retrieved snow depths are compared with the field surveys in Section 3, and with the modified climatology and ERA-Interim snowfall in Section 4. The last section concludes the paper with our recommendations for future approaches.

## 2     Data Description

In this section, we describe: (1) the snow radar and the steps followed in producing the echograms from which the air-snow (a-s) and snow-ice (s-i) interfaces are identified;  (2) the the data from two field programs; (3) the construction of snow depth estimates from climatology and from snowfall; and, (4) the five retrieval algorithms.

### 2.1     Snow radar

In the spring of 2009, an early version of the snow radar was installed and flown on the NASA P-3B to survey Arctic sea and land ice from Alaska and Greenland.  This version of the radar employed a fast-tuning, wideband voltage controlled oscillator (VCO) in a phase locked loop (PLL) configuration that used a FMCW chirp signal from a direct digital synthesizer (DDS) as a reference to produce a fast-sweeping, linear, and ultra-wideband chirp capable of collecting high-resolution sounding measurements from fixed wing aircraft (Panzer et al., 2013).  The measurements collected during these

initial surveys demonstrated a new capability to routinely measure snow depth over sea ice and annual snow layering over land ice. Since then, the system has been routinely deployed for both Arctic and Antarctic airborne campaigns as part of NASA OIB and to support the University of Kansas' Center for Remote Sensing of Ice Sheets (KU/CReSIS) and other collaborators' field programs (e.g., Patel et al., 2017; Yan et al., 2017). Below, we describe the system hardware and software processing algorithms used to generate the resulting data products and the changes that have been incorporated

throughout the evolution of the system to what is used today.

#### 2.1.1    Hardware

Since 2009, the snow radar hardware has been modified to improve system performance in terms of wider bandwidth, improved phase linearity at faster chirp rates, increased data acquisition rates and real-time hardware processing. Figure 1 shows the system block diagram, although specific values have changed throughout the various OIB deployments.

The basic components include a digital system, microwave chirp generator, microwave transmitter and receiver, and intermediate frequency (IF) receiver.  The digital system generates a 600-900 MHz reference chirp from the second Nyquist band of a 1 Giga-sample/s DDS, which is then multiplied by a factor of 20 by the PLL to generate a 12 to 18 GHz Ku-band





signal. The 12–18 GHz signal, which is typically used by the CReSIS Ku-band altimeter, is then down-converted using a mixer driven by a 10-GHz phase-locked oscillator to produce the system's 2–8 GHz transmit waveform. A directional coupler is used to replicate the chirp for the receiver. The receiver filters, amplifies and mixes the received waveform with the replicated chirp to produce an IF signal. This IF signal consists of a collection of near-constant "beat" frequency

components in which target delay is expressed by a simple relationship as the ratio of frequency by chirp rate. The spectral purity of each component is related to the frequency linearity of the 2-8 GHz chirp. Deviations from a linear frequency sweep result in reduced resolution and sidelobes. Finally, the sampled IF signal is coherently integrated or stacked 4 to 8 times.

### 2.1.2    Processing

The raw data products produced by the snow radar include a stream of coherently integrated (and optional Digital Down Converter - DDC) IF signals with respect to aircraft position.  These data are processed using the following steps:

1.  Locating IF Spectrum: The IF spectrum is found by taking the discrete Fourier transform of the raw data and mapping this result to time delay based on the Nyquist zone and DDC settings. The raw data are windowed to reduce range sidelobes. Additionally, any DDC induced modifications to coherent noise components must be compensated.

2.  Coherent noise removal: There are several unwanted coherent components within the system resulting in noise signals that vary slowly with respect to time.  These signals are estimated by analyzing the Doppler components of the data and then removed.

3.  Platform attitude and position corrections: Precision GPS and IMU information is used to position the antennas along the flight track. Relative altitude variations of the antennas within the coherent averaging window are corrected so that the

subsequent coherent averaging focuses the along-track beam pattern towards nadir.

4.  Windowing/Deconvolution: Phase and amplitude non-linearities of the system are estimated by analyzing data collected over specular returns produced by sea ice leads. A catalog of these responses are saved during a campaign and used to deconvolve non-linearities to produce low sidelobe waveforms. Figure 1 illustrates the effectiveness of the deconvolution process in suppressing sidelobes (noted in Kwok and Haas, 2015) for data collected during the 2012 OIB Arctic

campaign.

5.  Coherent/Incoherent Integration: Finally, additional coherent and incoherent integrations are used to improve signal-to-noise ratio (SNR), along-track resolution, and to reduce speckle.

### 2.2    Snow depth from field surveys

Coordinated surveys of field measurements and OIB overflights occurred in two years (2012 and 2014) under the

auspices of two different programs. Both were located on landfast ice to minimize to the variability introduced by the spatial



mismatch of the airborne and ground-based measurements due to ice drift. A brief description of these field programs is provided below.

### 2.2.1 BROMEX 2012

In coordination with OIB, *in situ* snow data were collected as part of the 2012 BROMEX near Barrow, Alaska
(Nghiem et al., 2013; Webster et al., 2014). Field measurements were conducted on smooth, first-year sea ice on Elson Lagoon, a location where landfast ice undergoes little deformation. Following the OIB pass on 15 March 2012, snow depths were measured every 1–5 m in a two-dimensional layout along two transects using an automated snow depth probe; the probe has an accuracy of 0.3 cm over level sea ice and snow (Sturm et al., 2006). The first transect (used here) consisted of three lines ~1000 m in length, each 5 m apart, for a width of 10 m. Snow density was measured every ~100 m with a Federal
Sampler (Marr, 1940); the average density was 306 kg m$^{-3}$ with a standard deviation of 91 kg m$^{-3}$. More information about the in situ data and field conditions is available in Webster et al. (2014).

### 2.2.2 Eureka 2014

During March and April of 2014 coordinated flight and a field-based campaigns (sponsored by ECCC) were carried out near Eureka, Nunavut, Canada to evaluate OIB estimates of snow depth on land-fast first-year ice (FYI). A predetermined
set of 11 parallel OIB flight lines were executed within Eureka Sound, a large inlet separating Axel Heiberg and Ellesmere Islands, on 25 March 2014. A spacing of 12 m between OIB flight lines ensured strong coincidence between the narrow swath of the snow-radar and the point-based nature of the planned field measurements. Snow depths within the radar footprints were sampled after the overflights. An extended description of the 2014 Eureka study site and field campaign can be found in (King et al., 2015).
As part of the field campaign, a linear transect was established to characterize local-scale variations in snow depth and density in proximity to the OIB snow radar footprint. Between the dates of 26 March 2015 and 29 March 2015 a total of 37,320 snow depth measurements were made along the transect covering a distance of 46 km. Measurements of snow depth were made with GPS enabled Snow-Hydro Magnaprobe units (Sturm et al., 2006), spaced by approximately 2 m between samples. In addition to measurements in the along-track flight direction, orthogonal transects of up to 100 m were completed
at random intervals to characterize variation in the radar across-track direction. Snow density was measured along the sampling transect at 174 locations with a coring device commonly referred to as the ESC-30 (Eastern Snow Conference 30 cm cross-section corer). Extracted cores were weighed *in situ* with a hanging scale to estimate bulk snow density and water equivalent.

Observed snow layer along the 2014 Eureka transect were shallow, with a mean snow depth of 17.8 cm and standard
deviation of 9.9 cm. This general condition corresponded with spatially predominance of large undeformed FYI pans (62% of ice regions under flight of the Eureka mission). Here, smooth ice topography and sustained Arctic winds allowed rapid redistribution of snow accumulation and limited mean depths to $16.0 \pm 8.3$ cm. Increased snow depth was associated with



local regions of deformed ice where drifted accumulation was found in proximity to convergence features (i.e. rafting, rubble, and pressure ridging). Those areas described as deformed FYI in King et al. (2015) and were shown to have a higher mean depth of $20.7 \pm 11.4$ cm. Density along the transect also varied in relation to the local ice topography with a mean of 306 kg m$^{-3}$ and standard deviation of 50 kg m$^{-3}$.

### 2.3    Snow depth from snowfall and climatology

Averaged retrievals (25 km × 25 km) from the five algorithms are compared with both snow depths from snowfall in ERA-Interim products and a modification of the W99 climatology. Below, the procedures for constructing these daily fields of basin-wide snow depths are described.

#### 2.3.1    Snow depth from ERA-Interim snowfall (ERAI-sf)

Fields of snow depth from ERA-interim snowfall are constructed following Kwok and Cunningham (2008). Daily snowfall on ice parcels (100 km × 100 km) across the Arctic Ocean is recorded on a daily basis. Ice drift is from optimally interpolated motion fields (described in Kwok et al., 2013). A daily cycle of accumulation and ice advection is carried out for each Lagrangian parcel, which mimics the process of snow accumulation over sea ice in motion. Starting from August 15, the snowfall on each drifting parcel is recorded and accumulated through the end of spring. Surface conditions (air temperature and ice concentration) determine when and where snow is allowed to accumulate. Accumulation is permitted only when the ERA-Interim 2-m air temperature is below freezing and the AMSR-E ice concentration exceeds 50%. When ice concentration drops below 50%, the accumulated snow is removed from that parcel. As ice concentrations rarely drop below 50% within the perennial ice pack, this condition is only relevant to the accumulation process over seasonal ice during the advance of the ice cover in the fall. Typically, the snow is thinner where the ice cover is formed later in the season (W99, Sturm et al., 2002a; Webster et al., 2014). The snow density climatology in Kwok and Cunningham (2008) (a modified version of that used in W99) is used to convert snow water equivalent into snow depth. No initial snow cover, representing snow that survived the melt season, is added to the multiyear ice at the beginning of the accumulation season. Henceforth, we refer to snow depth from this procedure as *ERAI-sf* estimates.

#### 2.3.2    Snow depth from modified climatology (modW99)

In this paper, we compute snow depth ($h_{fs}$) from the climatology (*W99*) following Kwok and Cunningham (2015) as,

$$h_s(X,t,f_{FY}) = h_s^W(X,t)(1-f_{FY}) + \alpha h_s^W(X,t)f_{FY} \ .$$

We use the annual space-varying snow depth and density, $h_s^W(X,t)$, from the snow climatology in W99. We note that this W99 climatology is from *in-situ* data collected between 1954 and 1991, and it is considered to be representative of snow depth over multiyear ice, so it does not address the snow depth over the seasonal ice cover of the Arctic Ocean. To account for reduced snow depth over FYI, we follow Laxon et al. (2013), who used a fraction ($\alpha$) of the climatological snow depth



to represent the reduced snow accumulation over FYI identified by Kurtz and Farrell (2011). Here $h_s$ is dependent on the fractional coverage of FYI ($f_{FY}$), derived from ASCAT scatterometer data (following Kwok, 2004). The $f_{FY}$ retrievals from ASCAT impart time-varying spatial patterns upon these otherwise static climatological fields. While Laxon et al. (2013) used a fixed value of $\alpha$=0.5, here we use $\alpha$=0.7, based on a subsequent analysis of CryoSat-2 ice thickness (Kwok

and Cunningham (2015). Since the above construction represents a modification of the W99 climatology, we henceforth refer to these snow depths as modW99 estimates.

### 2.4    Retrieval algorithms

Five sets of retrievals from five separate algorithms are considered here. The first algorithm below produced the standard products (2009–2013) that is archived at and distributed by the National Snow and Ice Data Center (NSIDC). The

remaining algorithms are devised by members on the Snow Thickness On Sea Ice Working Group (STOSIWG). Only brief summaries of each algorithm are provided here, and the reader is referred to the published literature for details on each of the algorithms. The retrieval algorithms differ in the way the air-snow (a-s) and snow-ice (s-i) interfaces are detected and localized in the radar returns. Figure 2 shows the different range locations of the interfaces in a collection of radar returns from Eureka; these differences are discussed in Section 3.5.

#### 2.4.1   Existing NSIDC product (NSIDC)

This algorithm was used for the Sea Ice Freeboard, Snow Depth, and Thickness data product that is archived at NSIDC with the following designation: IDSI4 (Kurtz et al., 2015). Details of the full algorithm methodology are described in (Kurtz et al., 2013). Briefly, the algorithm is an empirical method that selects the a-s interface using a combined peak and threshold method. The a-s interface is taken to be either the first significant peak above a defined threshold or the first point when the

rise in the radar return reaches a specified threshold if no peak is found. The method uses a linear scaling relation to the first version of the snow radar data collected in 2009 to select the thresholds used in the algorithm (as described in Kurtz and Farrell (2011)). The location of the s-i interface is defined as the maxima in the radar signal below the a-s interface.

#### 2.4.2   GSFC-NK

The GSFC-NK algorithm was used for the processing of quick look IceBridge data in 2014–2016

(https://nsidc.org/data/docs/daac/icebridge/evaluation_products/sea-ice-freeboard-snowdepth-thickness-quicklook-index.html), full details of the algorithm methodology are described in the product documentation at NSIDC. The algorithm is a waveform fitting method that follows the algorithm used for CryoSat-2 surface height retrievals described by Kurtz et al. (2014). A modification to that algorithm accounts for the different instrument characteristics of the snow radar and accounts for coherent scattering of the return using a heterogeneous flat patch surface model (Brown, 1982). The algorithm fits a

model waveform to the snow radar data using a bounded trust region method (Coleman and Li, 1996). Both the a-s and s-i interfaces are selected from the model fit results.





The algorithm is highly sensitive to the parameters used in the fitting process, and these are selected to provide a balance between the processing time needed and the quality of the fit obtained. The most important of these parameters are the initial guess and model fit bounds for the a-s and s-i interfaces along with the maximum number of iterations used in the fitting process. The initial guesses for the interface locations are taken from the empirical algorithm of the existing NSIDC

product, and the bounds for the possible interface locations are restricted to be within ±1.5 ns of the initial guesses. The maximum number of fit iterations was set to 300. Due to the large processing time needed for the algorithm, it was run on every other point for this comparison. The model fit quality was determined by calculating the sum of the waveform power divided by the squared norm of the fit residual, and fits with a value less than 0.2 were discarded.

### 2.4.3 SRLD

The Snow Radar Layer Detection (SRLD) algorithm was first developed for layer detection on land ice, in which the detection of the a-s interface and s-i interface are determined prior to the subsequent detection of deeper layers within the firn (Koenig et al., 2016). Over sea ice, only the a-s and s-i interfaces are returned by the algorithm. The initial application to existing OIB snow radar data from various campaigns (2009 - 2012) and the need for it to be applicable to future campaigns, required a process that would adapt to the data and not be dependent on fixed thresholds in the radar return signal. The

SRLD method uses the gradient between the open-air return values and the maximum return value to locate the two interfaces. The a-s interface is taken to be where the presence of snow has caused the radar return level to be elevated above the open-air values, near the beginning of the gradient. The s-i interface is taken to occur where the transition from snow to ice has produced a maximum peak in radar return level, at the end of the gradient. The open-air value is first determined by taking a sample median of the values above the surface for the data frame being analyzed. The median of the peak values is

used, along with the open-air value, to define the gradient to which a threshold is then applied for the determination of a-s interface. These gradient endpoints, comprised of the median air value and median peak value, adjust with the data from frame to frame. In the current implementation, the midpoint radar power level is used as the threshold. The point at which this radar power level maps onto each radar return profile determines the a-s interface, while the peak level in the return determines the s-i interface.

### 2.4.4 Wavelet

The Wavelet algorithm, described in Newman et al. (2014), operates on each trace (column) of an echogram independently and has three components: interface detection through the use of the Haar Wavelet Continuous Wavelet Transform (Haar-CWT), topographic filtering using the $h_{topo}$ parameter (to mitigate against heavily deformed ice topography on the radar point of closest approach), and the assignment of precision to each derived snow depth using radar system parameters.

The Haar-CWT is optimized for detecting abrupt transitions within a signal, such as those arising from interface returns, with the largest Haar-CWT coefficients localized at the largest magnitude signal transition. To detect the a-s and s-i interfaces, the echograms were preconditioned in different ways. To detect the a-s and s-i interfaces echograms are



preconditioned in different ways. To detect the a-s interface the logarithm of the echogram results in the largest magnitude signal transition occurring at the a-s interface. To detect the s-i interface the echogram is left in its original form, wherein the largest magnitude signal transition occurs at the s-i interface. The Haar-CWT is then applied to each echogram trace and the resulting coefficients are summed over a range of different scales: from the broad localization of the interface at large

wavelet scales to precise localization at small wavelet scales. The interface location is assigned to the location of the maximum of the summed coefficients. The benefits of this algorithm are that it is (1) robust and does not depend on a set of fixed thresholds and (2) the interface detection process is not affected by changes in transmitted power and receiver noise, which vary both during, and between, different OIB flight campaigns.

Interface picks associated with $h_{topo}$ values greater than 50 cm are ignored as they are associated with heavily

deformed ice topography, such as sea ice pressure ridges, where derived snow depths have been shown to be unreliable due to uncertainty in the radar scattering surface. Interface picks associated with $h_{topo}$ values of less than 50 cm are deemed valid and converted to snow depth by considering two-way travel time in the snow pack and the permittivity of snow.

Derived snow depths are assigned a precision based upon concurrent snow radar system parameters. Snow depth precision is calculated for each derived snow depth independently with the final precision value the sum of terms relating to

the SNR at the a-s interface, the signal-to-clutter ratio at the s-i interface, the fast-time range bin spacing, and the bandwidth-dependent range resolution of the snow radar.

### 2.4.5  JPL

This is a simplified version of a more involved algorithm by Kwok and Maksym (2014) that deals with residual system sidelobes in the returns. Since a re-processed version of the radar data set with suppressed system sidelobes (for all years

except 2013) is now available, this aspect of the algorithm has been disabled.

In this algorithm, both the s-i and a-s interfaces are detected and localized by determining the significance of each local maximum above the noise floor in individual echo returns. Significance is determined by the strength and width of the local maxima (power) and its associated leading/trailing edges relative to the expected noise power of the system. The system bandwidth controls the width (or sharpness) of a local maximum and the rate of rise of its leading edge. The algorithm uses

these system-dependent parameters to adapt to the changes in the radar system as the bandwidth and noise level of the snow radar have progressively improved over the course of the OIB mission. The highest significant peak in the echo profile is designated as the return from the s-i interface. Returns from a-s interfaces are assumed to be weaker and is the first significant range return, determined using the above criteria, above the s-i interface. Once the interfaces are detected, the radar range of the interface is localized in an oversampled (by 16 times) version of the echo return; this reduces the range

error in the identification of the local maxima in the echo return. From a scattering perspective, this restricts the detected returns from a-s interfaces that are more specular and appear as a detectable peak, rather than just as a strong leading edge.



## 3    Comparisons with field surveys

In this section, we first describe the comparison approach and the expected variability of the differences given the statistics of the retrievals and field measurements. Next, results from comparisons with measurements from BROMEX and Eureka are discussed. Lastly, we contrast the absolute range locations of the a-s and s-i interfaces from the different

algorithms, which provides insights into the preferred location on the echo profile that each algorithm designates as an interface. To account for the reduced propagation speed of the radar wave in the snow layer, all radar measurements below were converted to snow depth assuming a end-of-winter bulk density of 320 kg m$^{-3}$ (W99); snow depth estimates are relatively insensitive to uncertainties in bulk density (see error estimates in Kwok et al. (2011)).

### 3.1    Comparison approach

The spatial correlation length-scales (at $\rho$=0.5) of the point samples from both BROMEX and Eureka are short (~5–7 m; Figure 2). Hence, it is essential to select an averaging length scale that is more compatible with the coarser resolution of the snow-radar retrievals (5–10 m). We select an averaging radius of 20 m to allow for and to reduce the sensitivity of the comparisons to uncertainties in the snow-radar footprint, and to accommodate for uncertainties in the spatial overlap between the snow-radar footprint and the point samples from the field measurements. The number of *in-situ* snow depth

measurement in each averaged field samples for BROMEX and Eureka is 21 ± 5 and 15 ± 4, respectively. And, a 20-m radius represents an averaging of ~9 radar spots along-track, assuming a nominal spacing of ~5 m. The pre- and post-averaging spatial statistics of the field data are shown in Figure 2. For both datasets, the correlation length scale becomes broader (> 20 m) and variability is reduced. The resulting mean standard deviation ($\sigma_f$) of the BROMEX data within a 20-m circle reduced from 6.1 ± 1.5 to 2.2 ± 1.2 cm, and for the Eureka data from 7.1 ± 3.7 to 1.9 ± 1.6 cm.

To compare the averaged measurements (at a spacing 40 m), we take the difference between the estimates of from snow radar ($d_{sr}$) and field ($d_f$) as $\Delta d = d_{sr} - d_f$. Assuming that these two estimates are random variables that are uncorrelated and normally distributed, the variance of $\Delta d$ is expected to be $\sigma_{\Delta d}^2 = \sigma_{sr}^2 + \sigma_f^2$. The contribution of $\sigma_f$ is expected to be bounded by the values in the distribution shown in Figure 2, and the contribution of $\sigma_{sr}$ is discussed below. With the bounds on the two expected variances, $\sigma_{\Delta d}$ can be estimated for assessment of their differences.

### 3.2    Comparisons with snow depth from BROMEX

Figure 4 shows the comparison between four algorithms (NSIDC, GSFC-NK, SRLD, and JPL) with BROMEX field measurements. These results for the first transect are summarized in Table 2. The Wavelet retrievals were not available for this assessment. The OIB overflight of BROMEX covered a short distance of ~1000 m. Along this track, the field-measured snow depth (averaged) range between 15 and 30 cm. Except for the mean differences, it is apparent that the along-track

variability in snow depth is reproduced reasonably well, as measured by the correlations between the radar and field estimates (0.45–0.67). Of note is the lower scatter and reduced sample size in the distributions of $\sigma_{sr}$ for both the NSIDC



and GSFC-NK retrievals prior to averaging (Figure 4). This pattern can be attributed to the fact that both these algorithms used averaged waveforms in their retrievals, rather than raw waveforms provided in the radar data, and thus the data have already been smoothed and subsampled before the retrieval process. The averaging of the SRLD and JPL retrievals, in contrast, reduced $\sigma_{sr}$ by a factor of ~5. In any case, the standard deviation of the differences of ~3 − 4 cm (Figure 4) is

approximately what one would expect when calculating ($\sigma_{\Delta d}^2 = \sigma_{sr}^2 + \sigma_f^2$) using the values of $\sigma_f$ in Figure 2a and values of $\sigma_{sr}$ in Figure 4 (center panels). In these comparison, the mean differences vary between a maximum of –5 cm (GSFC-NK) and a minimum of +0.3 cm (NSIDC). These relative differences will be discussed in a Section 3.5.

### 3.3      Comparisons with snow depth from Eureka

The comparisons with Eureka field measurements are shown in Figure 5 and the results are summarized in Table 2.
This comparison includes many more samples from the multiple OIB tracks and more extensive sampling of the surface compared to the BROMEX survey. Averaged snow depth ranges from ~5 cm to more than 45 cm. This richer dataset tests the skill of five algorithms over a broad range of snow depths. Variability is between 0.29 and 0.72, as measured by the correlations between the radar and field estimates. The NSIDC and GSFC-NK algorithms have lower correlations, as their retrievals tend to underestimate snow depth; these two algorithms seem to be insensitive to snow for depths greater than ~20
cm in this case. The SRLD, Wavelet, and JPL retrievals have comparable correlations (0.66–0.72) with field data. Broadly, these algorithms tend to overestimate snow depths below ~10 cm, and with higher variability in retrievals over thicker snow (Figure 5). Except for the NSIDC retrievals, the standard deviation of the differences of ~4–5 cm is approximately as expected when calculated using the values of $\sigma_f$ in Figure 2b and values of $\sigma_{sr}$ ($\sigma_{\Delta d}^2 = \sigma_{sr}^2 + \sigma_f^2$; not shown here). In these comparison, the mean differences vary between a maximum of –6 cm (GSFC-NK) and a minimum of -1 cm (NSIDC).

We also examined the impact of surface roughness on the quality of the retrievals from the different algorithms (Table 2). We define roughness as $\sigma_f$, as defined above and as the standard deviation of the snow depth of the field samples used in creating the spatial average. If samples with roughness > 10 cm were not included in the comparisons, there is a consistent decrease in the standard deviation of the differences for all retrievals. This suggests that all the retrievals seem to be affected by surface roughness, consistent with the results reported by King et al. (2015). However, the correlation values did not
increase for all algorithms.

### 3.4      Algorithm-dependent filtering strategies: BROMEX and Eureka

For both the BROMEX and Eureka comparisons, it is clear that the different algorithms do not provide the same number of retrievals (Table 2) because of algorithm-dependent filtering strategies for removing low-quality estimates. Certainly, more conservative strategies reduce the retrieval rate, which may bias these comparisons. We address this concern
by examining the retrieval results for only those samples that are common to a pair of algorithms. Table 3 shows the results from all pairings of the five algorithms. Broadly, the changes in the comparison measures (i.e., $\rho$ and *Diff*) did not change



significantly for both BROMEX and Eureka. Compared to the diagonal element of the matrices in Table 3, the mean differences and standard deviations (relative to the field data) remain similar even when only the subset of common samples was used. Hence, we conclude that the statistics in Table 2 are fairly robust measures of the retrievals and relatively insensitive to the filtering strategies devised.

**3.5      Different approaches to localize the air-snow and snow-ice interfaces**

As observed earlier, the comparisons show that the biases (or mean differences) for individual algorithms are consistent (similar in magnitude) at both BROMEX and Eureka.  For example, the mean differences of the SRLD and JPL retrievals are approximately +2 cm and –2 cm, respectively, at both field sites (see Table 2). Thus, on average, the SRLD snow depth retrievals are 4 cm higher than the JPL retrievals. These patterns suggest that systematic biases exist within individual

algorithms due to their range determinations of the a-s and s-i interfaces in the echograms. In each retrieval algorithm, a particular characteristic of a leading edge (see Figure 3b) or return peak is typically used to determine the range point to be the location of an interface. In this case, the same characteristic has to be used consistently used for both the a-s and s-i interfaces, otherwise the range distance between the two interfaces will be biased.

Here, we compare the range locations of the two interfaces selected by four of the five algorithms to assess the

contribution of algorithm-induced biases seen in the observed mean differences at the Eureka site (Figure 6). The left panel (Figure 6) shows whether there are mean differences in the retrieved range distances (calculated from the range locations of the two identified interfaces) from two algorithms. We first discuss the comparison of the SRLD and JPL retrievals (Figure 6f) since this is the simplest case, where the range distance calculated by the SRLD algorithm is always greater than that obtained by JPL. Here both algorithms use the peak in a given echogram as the location of the s-i interface, so they have a

narrow distribution of differences in the range location of that interface (~1 cm). Nearly all of their difference in retrieved snow depth can be attributed to differences in the location of the a-s interfaces (Figure 6f). The SRLD algorithm locates the a-s interface on the leading edge, while the JPL algorithm located it at the local peak following the leading edge. Relative to the JPL a-s interfaces, the SRLD a-s interfaces are always displaced toward the radar, i.e., in the near range, therefore shorter range. Thus, the range distance to the JPL a-s interface is always higher. The same arguments apply to explaining some of

systematic differences in the range location observed in the other comparisons shown in Figure 6. This pattern suggests that systematic choices made in the localization of the interfaces in each algorithm largely explain the consistent mean inter-algorithm differences (Table 2). A related question is where on the echo profile one should pick as the location of an interface this will not be addressed here.

**4      Basin-scale assessments**

We next assess the retrieved OIB snow depths from all Arctic campaigns at a longer averaging length-scale (> 10 km). First, we summarize the overall retrievals from four algorithms available for this analysis (NSIDC, GSFC-NK, SRLD and JPL). Second, the consistency of retrievals at crossovers and repeat tracks (12.5 km) is examined. Last, we compare the averaged retrievals (at 25 km) with the fields of snow depth constructed from ERA-Interim snowfall (ERAI-sf) and modified



climatology (modW99). These large-scale comparisons provide a broad assessment of the spatial and interannual variability of the snow radar retrievals, along with their relative agreement with the two reconstructed fields of snow depth.

### 4.1 Summary of basin-scale snow depths: 2009-2015

Figure 7 show seven years (2009–2015) of retrievals (25-km averages) from four algorithms NSIDC, GSFC-NK, SRLD, and JPL). These maps show the spatial differences in the retrievals from all the OIB flight tracks for a particular Arctic campaign. Figure 8 shows the associated snow depth distributions in three multiyear/seasonal ice regimes (MYf $\geq 0.7$; $0.3 < $ MYf $< 0.7$; MYf $\leq 0.3$), where MYf is the fraction of multiyear sea ice MYI) coverage within each 25-km sample. The calculated mean and standard deviation of the sample population of snow depth in each of the three categories are shown on the top left corner of each panel.

As seen in the spatial maps and distributions (Figures 7 and 8), the snow depth within the MYI cover north of Greenland and next to the Canadian Arctic Archipelago is highest in all the retrievals. In general, the average snow depth over the seasonal ice is thinner than those within the MYI cover. Of note is that the snow depth in the southern Beaufort Sea is always the thinnest, especially for those tracks between 2012 and 2015. The differences in retrievals from the four algorithms are examined in more detail in Section 4.3.

### 4.2 Comparisons at crossovers and repeat tracks

Here we examine the consistency of the snow radar estimates (12.5 km averages) at available crossovers of the OIB flight tracks over the seven years of available data. Figure 9a shows the retrieval crossover differences (up to 164) and the time separation between them. When the time separations are short (<10 days), the differences are at the centimeter level and quite consistent (high correlation) for all algorithms. Even with the expected spatial variability, these differences remain stable over the seven years and thus indicate the general consistency of individual retrieval algorithms. Crossover consistency also suggests a relatively long, isotropic spatial correlation length scale for snow depth. That is, at longer length scales snow depth varies slowly spatially and is likely due to synoptic scale patterns. As expected, when the time separation increases, so do the differences. At separations of more than twenty days, the differences are generally higher and positive, which seems consistent with the initial expectation that changes could be attributed to snowfall. However, differences are also likely due to advection of ice parcels with thicker snow covers into the crossover point. As most of the crossovers are located in regions with fairly large spatial gradients in snow depth, attribution of the observed differences is difficult.

Figure 9b compares the snow depth at four available repeat tracks. These near-exact repeat tracks are of outbound and return segments flown during the same flight, so the time separations are typically less than a few hours. Except for ice drift, the snow radar should be acquiring data over similar snow and ice conditions. The difference distributions show that the mean differences are at most ~2–3 cm. The sign and magnitude of the differences are consistent across all the retrievals and suggest that these are valid mean differences in the estimated snow depth. These comparisons also suggest that even though the algorithms produce self-consistent results, the mean snow depth can be quite different between the algorithms, as is evident in the difference in retrieved snow depths from the four algorithms (Figure 9b).



### 4.3 Comparisons with snow depth from ERA-Interim snowfall (ERAI-sf) and modified climatology *(modW99)*

The snow depth estimates from ERAI-sf, modW99 and their differences along OIB flight tracks are shown in Figure 10a. The large-scale patterns are similar with thinner snow over seasonal ice and thicker snow in regions with higher MYf, especially north of Greenland and the Canadian Arctic Archipelago. Relative to the modW99 estimates, the ERAI-sf snow

depths are broadly lower over the entire Arctic Ocean ice cover, except in 2012.

Figure 11 shows the relative inter-annual variability (IAV) of the mean snow depth of the two fields over seven years for the same three sea ice categories. As expected, the IAV of modW99 fields are lower than those from ERAI-sf, because the modW99 estimates are a static monthly climatology (W99) and are modulated by only the timing of the OIB flights and spatial variability of MYf derived from satellite data. The largest differences can be seen in the southern Beaufort Sea in

2014 and 2015, where ERAI-sf snow depths are lower than those from modW99. This difference can be largely attributed of the presence of a tongue of MYI that advected into this region, which was used in the construction of the modW99 fields but is absent in the ERAI-sf fields (because the MYf is not used in the estimates). Away from the coastal zones and away from the transition zone between the MYI and seasonal ice cover, these differences are less pronounced.

Figure 11 also shows the IAV in the mean snow depth from the four algorithms. Three of the algorithms (NSIDC, GSFC-

NK, and SRLD) have relatively large IAV compared to the two constructed fields. In the mean, the NSIDC snow depths are lower than the constructed fields. The large increase/decrease in the GSFC-NK retrievals in 2012/2013 is much larger than the expected IAV of ~6 cm between March and April (W99). Over the seven years, the SRLD retrievals show significant snow depths trend in all three categories of MYI coverage, which seems unrealistic. The IAV of the JPL retrievals are more similar to those of the constructed fields.

Figures 12 and 13 show the spatial comparisons with the constructed fields. Variability in the differences between the retrievals and ERAI-sf and modW99 is expected. The mean snow depth from the snow radar represents the average of all age types within a track, whereas the ERAI-sf and modW99 estimates represent the accumulated snow depth since the beginning of the season with no consideration of wind-driven redistribution, loss into leads, or the introduction of seasonal ice of variable age (due to deformation of the ice cover). If any of those factors were significant for a given year, it would

change the mean snow depth observed by the snow radar. Broadly, the IAV of the retrievals for the different ice categories in Figure 11 is consistent with the variability in the difference maps (Figures 12 and 13).

The contrast in IAV in retrievals from the four algorithms underscores the importance of multi-year assessments. The large IAV of retrievals from a given algorithm suggest issues in algorithmic robustness in adapting to changes in radar data quality. Since the snow radar has changed and improved over time (Table 1), the retrieval algorithms must adapt to

changes in data quality. It is important to note that even though the retrievals may be internally consistent for a given algorithm, as suggested in the Section 4.2, whether algorithms are sensitive to changes in radar data quality or radar hardware parameters depends on the specific characteristics of each algorithm.





## 5 Conclusions

In this study, we compared the retrievals from five different algorithms with each other and with snow depth measurements from two field surveys, modified climatology, and snow depths derived from ERA-Interim products. The inter-comparisons amongst different retrievals and comparisons with field measurements allow a detailed assessment of the retrievals, while the comparisons with climatology and analyzed snowfall provide a broader multi-year perspective of their inter-annual retrieval consistency and robustness to changes on radar parameters, and their relative agreements with basin-scale fields. We reiterate that the aim of the work was not to the select the best algorithm, but rather to provide results that would serve to inform the development of the next-generation retrieval algorithm. Below we highlight the salient points from the above inter-comparisons, and important considerations in the development and assessment of retrieval algorithms:

- When comparing snow-radar retrievals with point snow depth samples, it is essential to select an appropriate length scale such that the differences in the comparisons are not dominated by the geophysical variability. Hence, the fairly large number of point samples within a radar footprint is crucial in generating the averaged field dataset.

- Comparisons with BROMEX and Eureka field data show that, even though there are residual biases, the profile of along-track snow depth can be reproduced and the sensitivity of the snow radar to the a-s and s-i interfaces in the varied conditions at the two field sites can be evaluated.

- Retrieval algorithms differ in the way the a-s and s-i interfaces are detected and localized in the radar returns, and algorithm choices made in the localization of the interfaces by each algorithm mostly explain the consistent mean differences seen in the inter-comparisons and in the comparisons to field data. A related and important question regarding where on the echo profile one should pick as the location of an interface is not addressed here.

- Most of the retrievals were able to reproduce the expected spatial pattern of higher snow depth in MYI regions north of Greenland and near to the Canadian Arctic Archipelago, along with the thinner sea ice cover toward the Beaufort Sea to the southwest.

- The examination of retrievals at cross-overs and repeat tracks shows internal consistency of individual retrieval algorithms. These comparisons also suggest that, even though the algorithms may produce self-consistent results, the mean snow depth may be very different between algorithms, so this metric cannot be used as a measure of the overall performance of a retrieval approach

- Over the seven years, several algorithms have relatively large interannual variability that is higher than expected by climatology (W99). Large interannual variability of retrievals from a given algorithm suggests issues in algorithmic robustness in adapting to changes in radar data quality. The contrast in interannual variability in retrievals from the four algorithms underscores the importance of multi-year assessments, especially when the instrument performance have changed and improved, as is the case here. Of note is that 2009 snow radar had lower SNR, and the processed 2013 data had large, persistent sidelobes that affected the retrieval algorithms. To provide a long-term, consistent record of snow



depth for understanding changes, algorithms with set thresholds may need to be tuned appropriately for the duration of the record.

*Acknowledgments.* We thank S. S. Pang for her software support during the course of this work. RK carried out this work at the Jet Propulsion Laboratory, California Institute of Technology, under contract with the National Aeronautics and Space Administration. Development and testing of the Wavelet algorithm was supported by Office of Naval Research award N000141210512 and the NOAA Ocean Remote Sensing program.





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



**Table 1**
Radar parameter used in Arctic surveys

| Field Season | 2009 | 2010–2011 | 2012–2015 |
|---|---|---|---|
| Frequency Range (GHz) | 2.5–7.0 | 2.0–6.5 | 2.0–8.0 |
| Pulse Length (µs) | 270 | 250 | 250 |
| Transmit Power (mW) | 10 | 20-100 | 100 |
| IF Frequency Range (MHz) | 29.2–58.32 | 31.25–62.50 | 62.50–125.00 |
| Sampling Rate (Msamples/s) | 58.32 | 62.5 | 125 |



**Table 2**

Comparison of averaged radar snow depth with field measurements at BROMEX and Eureka. The additional comparisons at
5   Eureka are for samples where the standard deviation of the field-measured snow depth within the averaging radius of 20 m is
less than 10 cm.

| (cm) | | NSIDC | GSFC-NK | SLRD | Wavelet | JPL |
|---|---|---|---|---|---|---|
| BROMEX | $\rho$ | 0.54 | 0.44 | 0.66 | | 0.67 |
| | $N$ | 227 | 113 | 227 | | 169 |
| | *Diff* | 0.3±4.17 | −4.8±4.0 | 4.2±3.4 | | −1.8±3.4 |
| Eureka | $\rho$ | 0.29 | 0.45 | 0.66 | 0.72 | 0.63 |
| | $N$ | 4666 | 1746 | 4479 | 3042 | 3069 |
| | *Diff* | −0.8±11.2 | −5.7±5.1 | 1.3±4.5 | 2.0±4.7 | −2.2±4.6 |
| $\sigma < 10$ | $\rho$ | 0.34 | 0.61 | 0.71 | 0.67 | 0.60 |
| | $N$ | 3998 | 1477 | 3905 | 2631 | 2642 |
| | *Diff* | −0.1±9.9 | −4.5±2.4 | 1.8±3.6 | 1.8±4.3 | −2.3±3.7 |



**Table 3**

Comparison of radar snow depth with field measurements at BROMEX and Eureka using retrievals at locations that are common to both algorithms. In the matrix presentation below, the snow depth retrieved by the row algorithm (at locations common with the column algorithm) are used in the comparisons with field data. Note that the matrix is not symmetric.

| (*Diff* in cm) | | NSIDC | GSFCNK | SLRD | Wavelet | JPL |
|---|---|---|---|---|---|---|
| **BROMEX** | | | | | | |
| NSIDC | $\rho$ | 0.54 | 0.52 | 0.54 | | 0.70 |
| | $N$ | 227 | 113 | 227 | | 169 |
| | *Diff* | 0.3±4.2 | 0.5±4.2 | 0.3±4.2 | | 0.8±3.0 |
| GSFC-NK | $\rho$ | 0.44 | 0.44 | 0.44 | | 0.39 |
| | $N$ | 113 | 113 | 113 | | 80 |
| | *Diff* | −4.8±4.0 | −4.8±4.0 | -4.8±4.4 | | -4.9±4.5 |
| SRDL | $\rho$ | 0.67 | 0.75 | 0.66 | | 0.68 |
| | $N$ | 227 | 113 | 227 | | 169 |
| | *Diff* | 4.2±3.4 | 4.6±3.1 | 4.2±3.4 | | 5.2±3.2 |
| JPL | $\rho$ | 0.67 | 0.60 | 0.67 | | 0.67 |
| | $N$ | 169 | 80 | 169 | | 169 |
| | *Diff* | −1.8±3.4 | −1.2±3.8 | −1.8±3.4 | | −1.8±3.4 |
| **Eureka** | | | | | | |
| NSIDC | $\rho$ | 0.29 | 0.45 | 0.29 | 0.25 | 0.28 |
| | $N$ | 4666 | 1745 | 4448 | 3023 | 3059 |
| | *Diff* | −0.8±11.2 | −2.2±5.3 | −0.7±11.0 | −0.3±12.6 | −0.9±9.5 |
| GSFC-NK | $\rho$ | 0.44 | 0.45 | 0.48 | 0.50 | 0.46 |
| | $N$ | 1745 | 1746 | 1699 | 1119 | 820 |
| | *Diff* | −5.7±5.1 | −5.7±5.1 | −5.5±4.7 | −5.4±4.7 | −5.0±4.2 |
| SRDL | $\rho$ | 0.66 | 0.64 | 0.66 | 0.69 | 0.59 |
| | $N$ | 1112 | 1699 | 4479 | 2991 | 3003 |
| | *Diff* | 1.4±4.4 | 1.5±4.8 | 1.3±4.5 | 1.5±4.3 | 1.8±4.1 |
| Wavelet | $\rho$ | 0.72 | 0.67 | 0.71 | 0.72 | 0.67 |
| | $N$ | 3032 | 1119 | 2991 | 3042 | 2016 |
| | *Diff* | 2.0±4.7 | 2.0±5.1 | 1.8±4.5 | 2.0±4.7 | 2.7±4.4 |
| JPL | $\rho$ | 0.63 | 0.58 | 0.61 | 0.57 | 0.63 |
| | $N$ | 3059 | 820 | 3003 | 2016 | 3069 |
| | *Diff* | −2.2±4.6 | −2.3±4.3 | −2.4±4.4 | −2.0±4.4 | −2.2±4.6 |



## Figure Captions

Figure 1. (a) Snow-radar hardware block diagram. Strength and location of sidelobes in the system impulse response (b) before and (c) after deconvolution of the radar data acquired during 2012 OIB Arctic campaign. $w$ is the half-width of the mainlobe. Family of curves shows dependence on peak-signal-to-noise ratio (*PSNR* in dB) and is constructed using all the radar echos ($\sim 4 \times 10^6$) from the campaign. Normalized returns are for every 1-dB increment of *PSNR* between 10 and 50 dB.

Figure 2. Location of the air-snow and snow-ice interfaces overlaid on power of return echoes. (a) Examples from four algorithms (black trace – air-snow interface; blue trace – snow/ice). (b) Except for the modeled GSFC-NK algorithm, the variability in the interface locations (e.g., leading edge, peak, etc.) depends on the algorithm.

Figure 3. Spatial correlation (left and center) and variability (right panel) of snow depth measurements from (a) BROMEX and (b) Eureka before (black lines) and after (green lines) averaging (diameter of 40 m). Gray bars show the number of measurements used in each correlation calculations.

Figure 4. Comparison of four algorithms with field measurements from BROMEX. (a) NSIDC (b) GSFC-NK. (c) SRLD. (d) JPL. (Left panel) Averaged field (black dots) and retrieved (red dots) snow depth with standard deviation at each sample (gray band). (Center) Standard deviation of retrieved snow depth before (black) and after (green) averaging – 40 m along track. (Right) Scatterplot shows correlation (R) between the averaged snow depth samples, the number of samples, and the mean/standard deviation of the differences are show on the top left corner of plot.

Figure 5. Comparison of five algorithms with field measurements from Eureka. (a) NSIDC. (b) GSFC-NK. (c) SRLD. (d) Wavelet. (e) JPL. Scatterplot shows correlation (R) between the averaged snow depth samples, the number of samples, and the mean/standard deviation of the differences are show on the top left corner of plot.

Figure 6 Distribution of differences in range locations. (a) GSFC-NK minus Wavelet. (b) GSFC-NK minus JPL. (c) GSFC-NK minus SRLD. (d) Wavelet minus SRLD. (e) Wavelet minus JPL. (f) SRLD minus JPL. (Left panel) differences in the range between the two interfaces, (center panel) differences in range to the snow-ice (s-i) interface, and (right panel) differences in range to the air-snow (a-s) interface.

Figure 7. Seven years (2009-2015) of retrieved snow depths from four algorithms. (a) NSIDC. (b) GSFC-NK. (c) SRLD. (d) JPL. Snow depths (top left corner) are averages based on three categories of multiyear-ice fractions (MYf) shown in legend. Background shows multiyear sea ice fraction (dark gray: MYf $\geq 0.7$; light gray: $0.3 < MYf < 0.7$; white: MYf $\leq 0.3$). Corresponding snow depth distributions are shown in Figure 8. The gray shadings represent regions with $>0.7$ (darker) and $>0.9$ (lighter) MYf.

Figure 8. Snow depth distributions in seven years (2009-2015) of retrievals (a) NSIDC. (b) GSFC-NK. (c) SRLD. (d) JPL. Quantities on top right corners are mean, standard deviation, of the sample populations in three multiyear/seasonal ice regimes (MYf $\geq 0.7$; $0.3 < MYf < 0.7$; MYf $\leq 0.3$).



Figure 9. Differences of snow depth retrievals at cross-overs and repeat tracks. (a) Cross-overs with varying time separation. Mean/standard deviation of differences and correlation coefficients (R) are shown on each plot. (b) Repeat tracks: 2 April 2010, 5 April 2010, 17 March 2011, and 22 March 2012. Inset shows the distribution of differences and quantities show mean/standard deviation of differences, correlation, and sample population. Each sample represents the

5      mean snow depth in a 4-km along track segment. (Note: On 21 April 2009, the snow radar was operated with an altitude difference of ~300 m in the outbound and return tracks.).

Figure 10. Comparison of snow depth from ERAI-sf and modW99 climatology at the snow radar tracks. (a) ERAI-sf; (b) modW99; and, (c) Difference between (a) and (b). The mean and standard deviation in each of the three categories of MYf are shown on the top left corner of each panel.

10   Figure 11. Comparison of snow depth retrievals from four algorithms (NSIDC, GSFC-NK, SRLD, JPL) with ERAI-sf and modW99 estimates in three multiyear/seasonal ice regimes (MYf ≥0.7; 0.3 < MYf < 0.7; MYf ≤ 0.3).

Figure 12. Comparison of snow depth retrievals with ERAI-sf and the modW99 estimates. (a) NSIDC. (b) GSFC-NK.

Figure 13. Comparison of snow depth retrievals with ERA-I snowfall and modW99. (a) SRLD. (b) JPL.



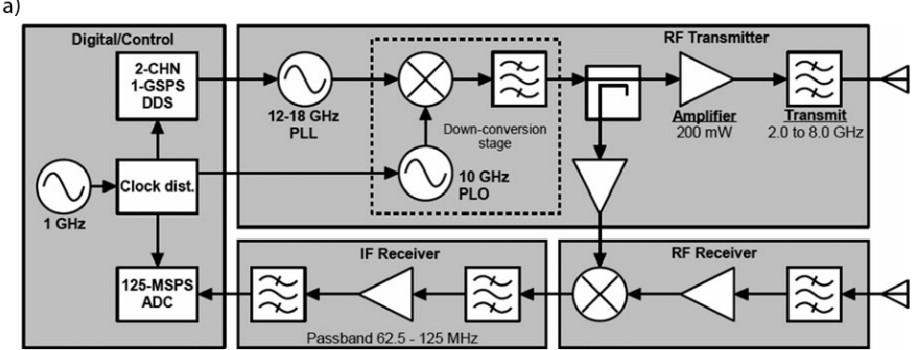

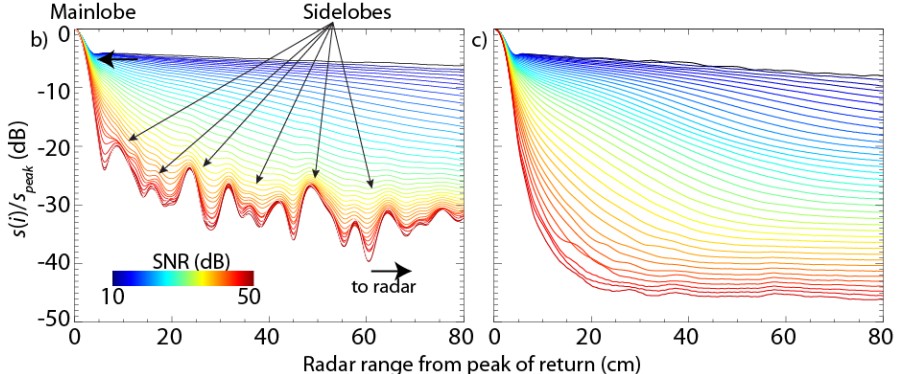

**Figure 1. (a) Snow-radar hardware block diagram. Strength and location of sidelobes in the system impulse response (b) before and (c) after deconvolution of the radar data acquired during 2012 OIB Arctic campaign. $w$ is the half-width of the mainlobe. Family of curves shows dependence on peak-signal-to-noise ratio ($PSNR$ in dB) and is constructed using all the radar echos (~4×10$^6$) from the campaign. Normalized returns are for every 1-dB increment of $PSNR$ between 10 and 50 dB.**



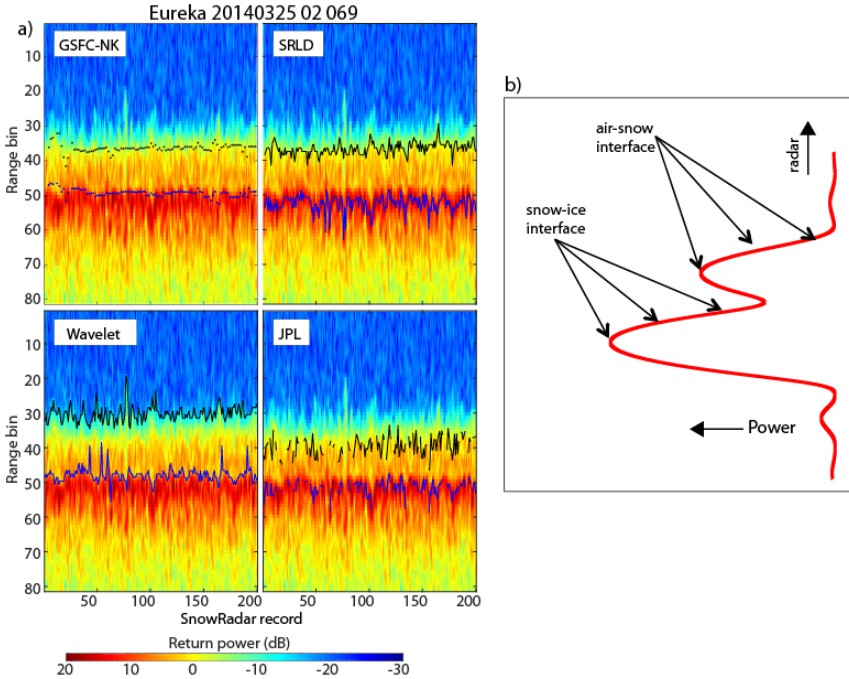

**Figure 2.** Location of the air-snow and snow-ice interfaces overlaid on power of return echoes. (a) Examples from four algorithms (black trace – air-snow interface; blue trace – snow/ice). (b) Except for the modeled GSFC-NK algorithm, the variability in the interface locations (e.g., leading edge, peak, etc.) depends on the algorithm.





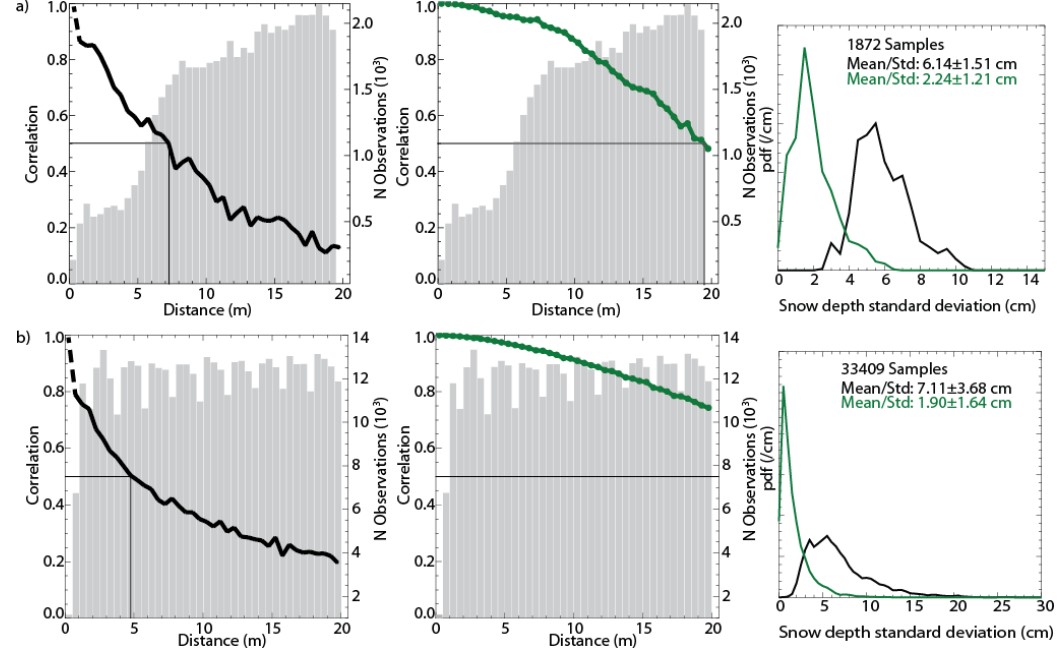

**Figure 3. Spatial correlation (left and center) and variability (right panel) of snow depth measurements from (a) BROMEX and (b) Eureka before (black lines) and after (green lines) averaging (diameter of 40 m). Gray bars show the number of measurements used in each correlation calculations.**





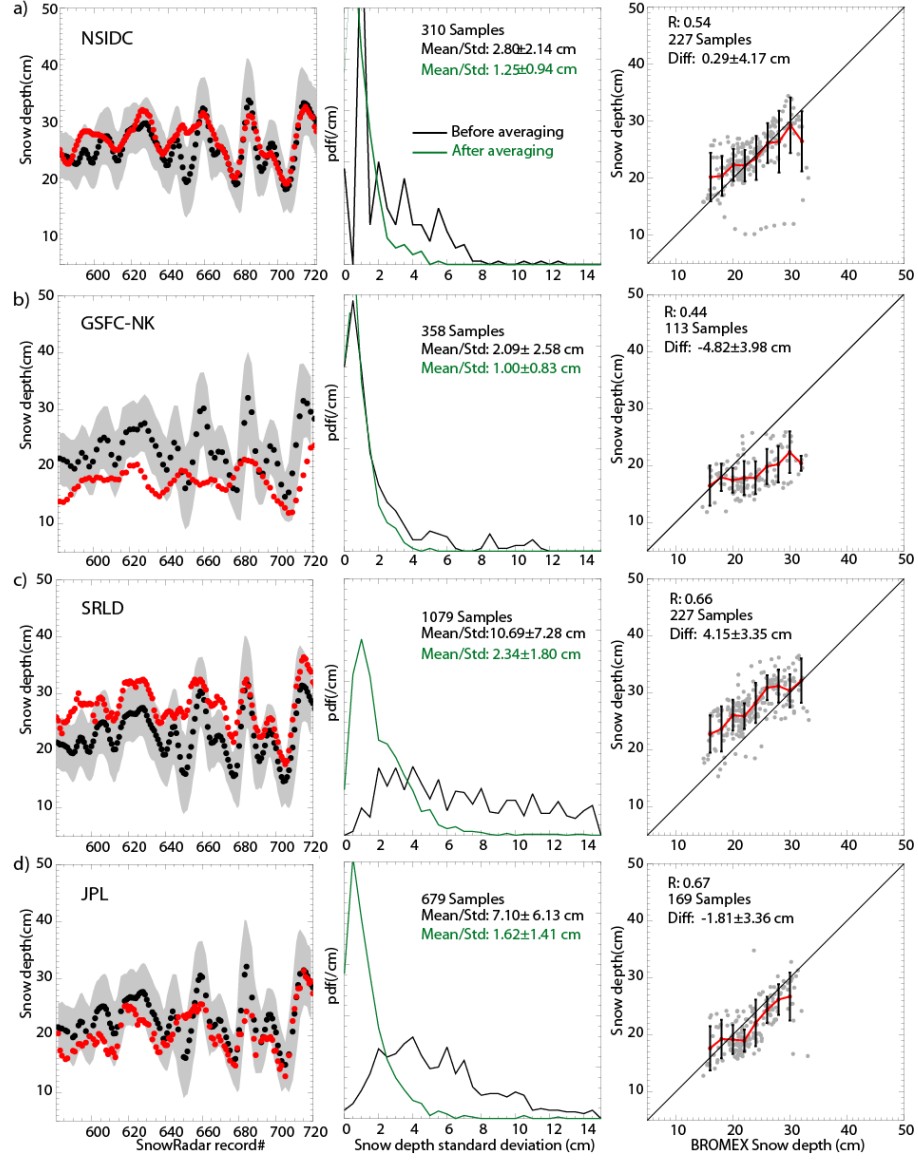

**Figure 4. Comparison of four algorithms with field measurements from BROMEX. (a) NSIDC. (b) GSFC-NK. (c) SRLD. (d) JPL. (Left panel) Averaged field (black dots) and retrieved (red dots) snow depth with standard deviation at each sample (gray band). (Center) Standard deviation of retrieved snow depth before (black) and after (green) averaging – 40 m along track. (Right) Scatterplot shows correlation (R) between the averaged snow depth samples, the number of samples, and the mean/standard deviation of the differences are show on the top left corner of plot.**



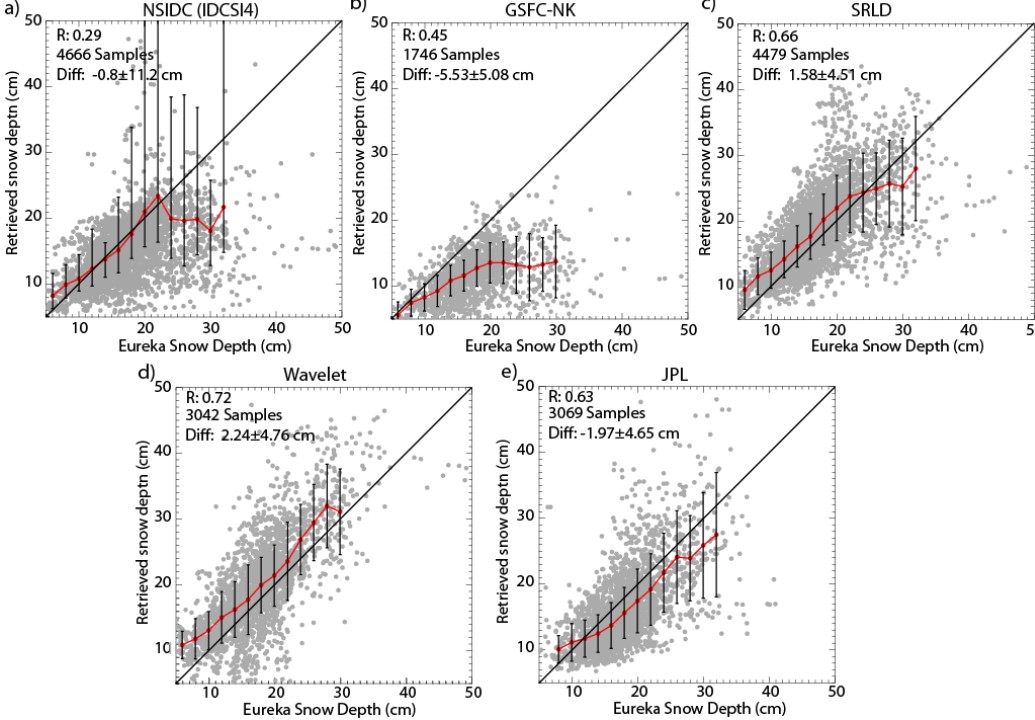

**Figure 5. Comparison of five algorithms with field measurements from Eureka. (a) NSIDC. (b) GSFC-NK. (c) SRLD. (d) WAVELET. (e) JPL. Scatterplot shows correlation (R) between the averaged snow depth samples, the number of samples, and the mean/standard deviation of the differences are show on the top left corner of plot.**



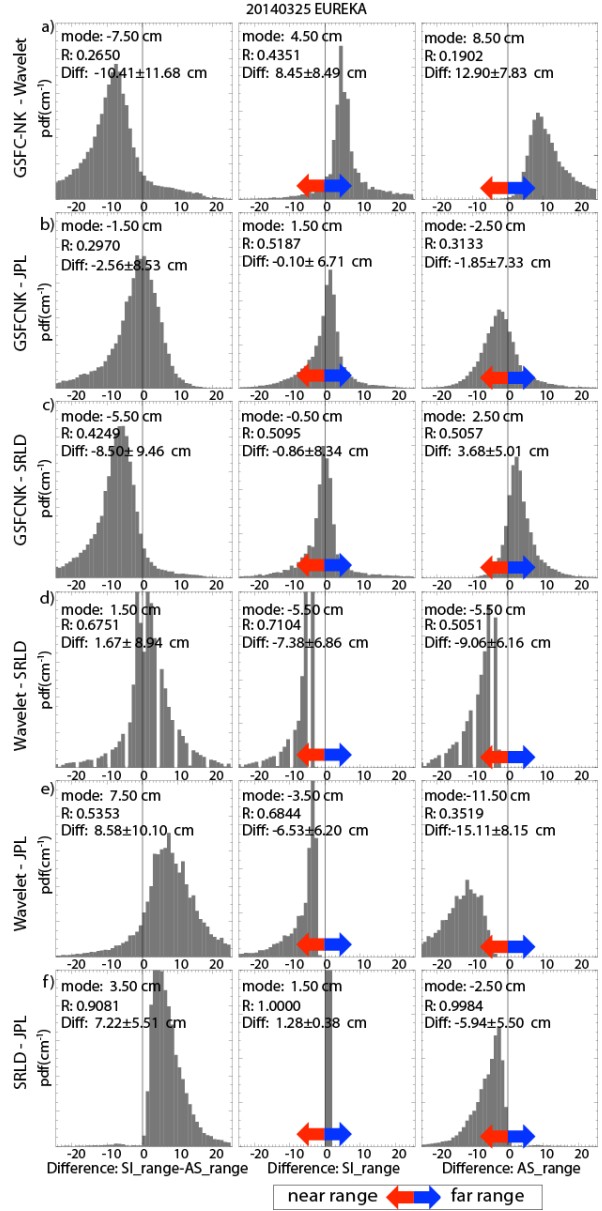

**Figure 6 Distributions of differences in range locations. (a) GSFC-NK minus Wavelet. (b) GSFC-NK minus JPL. (c) GSFC-NK minus SRLD. (d) Wavelet minus SRLD. (e) Wavelet minus JPL. (f) SRLD minus JPL. (Left panel) differences in the range between the two interfaces, (center panel) differences in range to the snow-ice (SI) interface, and (right panel) differences in range to the air-snow (AS) interface.**



**Figure 7. Seven years (2009-2015) of retrieved snow depths from the four algorithms. (a) NSIDC. (b) GSFC-NK. (c) SRLD. ((d) JPL. Snow depths (top left corner) are averages based on three categories of multiyear-ice fractions (MYf) shown in legend. Background shows multiyear sea ice fraction (dark gray: MYf ≥0.7; light gray: 0.3 < MYf < 0.7; white: MYf ≤ 0.3). Corresponding snow depth distributions are shown in Figure 8. The gray shadings represent regions with >0.7 (darker) and >0.9 (lighter) MYf.**

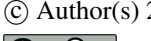


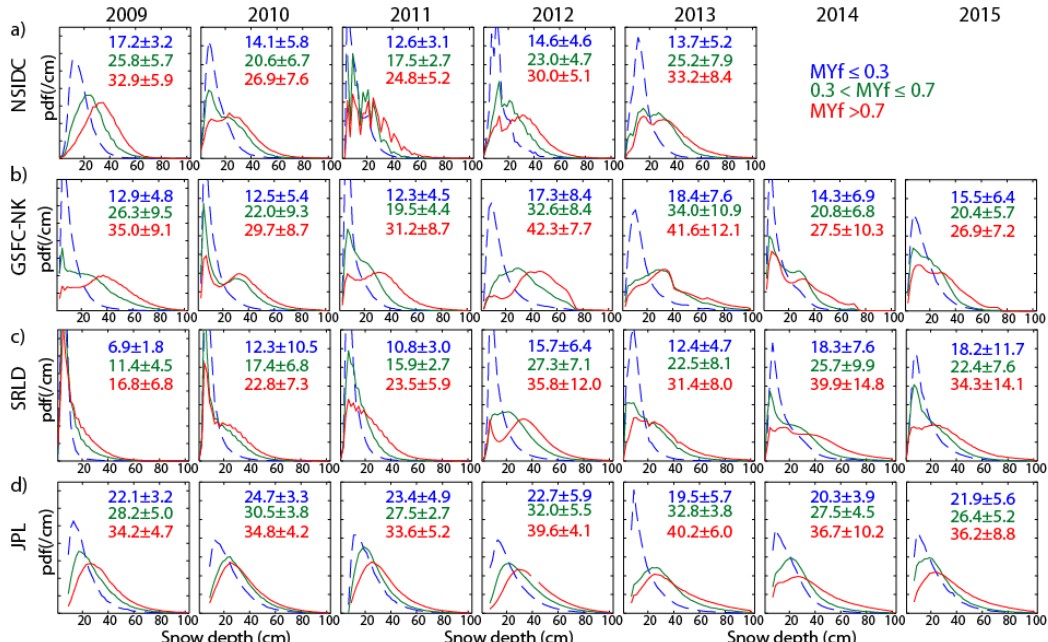

**Figure 8.** Snow depth distributions in seven years (2009-2015) of retrievals (a) NSIDC. (b) GSFC-NK. (c) SRLD. (d) JPL. Quantities on top right corners are mean, standard deviation, of the sample populations in three multiyear/seasonal ice regimes (MYf ≥0.7; 0.3 < MYf < 0.7; MYf ≤ 0.3).



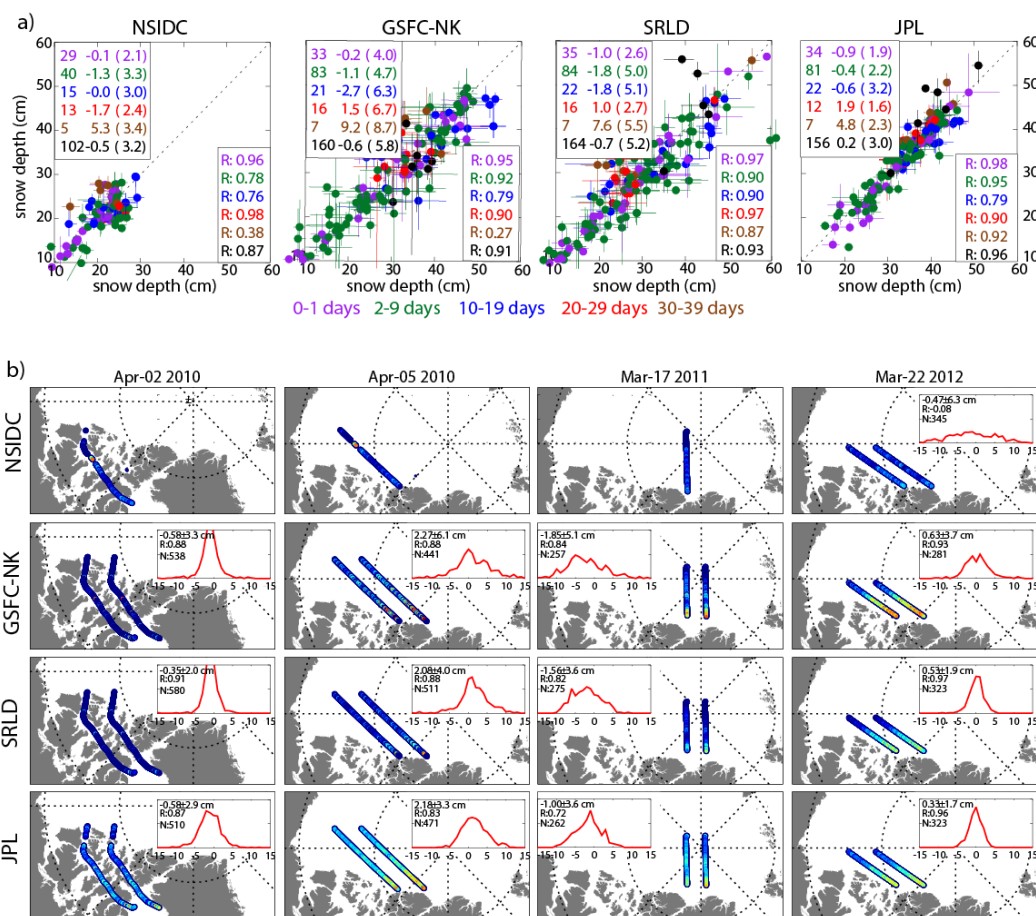

**Figure 9. Differences of snow depth retrievals at cross-overs and repeat tracks. (a) Cross-overs with varying time separation. Mean/standard deviation of differences and correlation coefficients (R) are shown on each plot. (b) Repeat tracks: 2 April 2010, 5 April 2010, 17 March 2011, and 22 March 2012. Inset shows the distribution of differences and quantities show mean/standard deviation of differences, correlation, and sample population. Each sample represents the mean snow depth in a 4-km along track segment. (Note: On 21 April 2009, the snow radar was operated with an altitude difference of ~300 m in the outbound and return tracks.).**





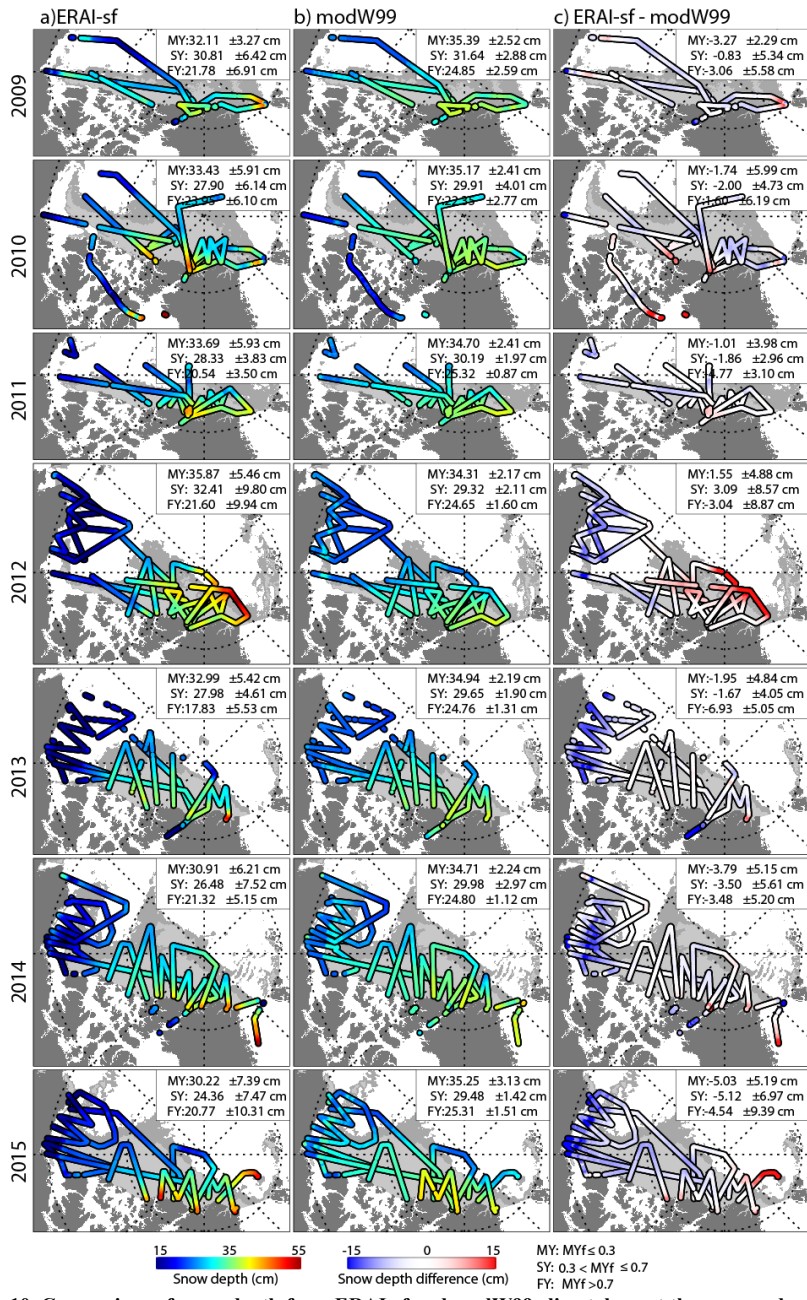

**Figure 10. Comparison of snow depth from ERAI-sf and modW99 climatology at the snow radar tracks. (a) ERAI-sf; (b) modW99; and, (c) Difference between (a) and (b). The mean and standard deviation in each of the three categories of MYf are shown on the top left corner of each panel.**



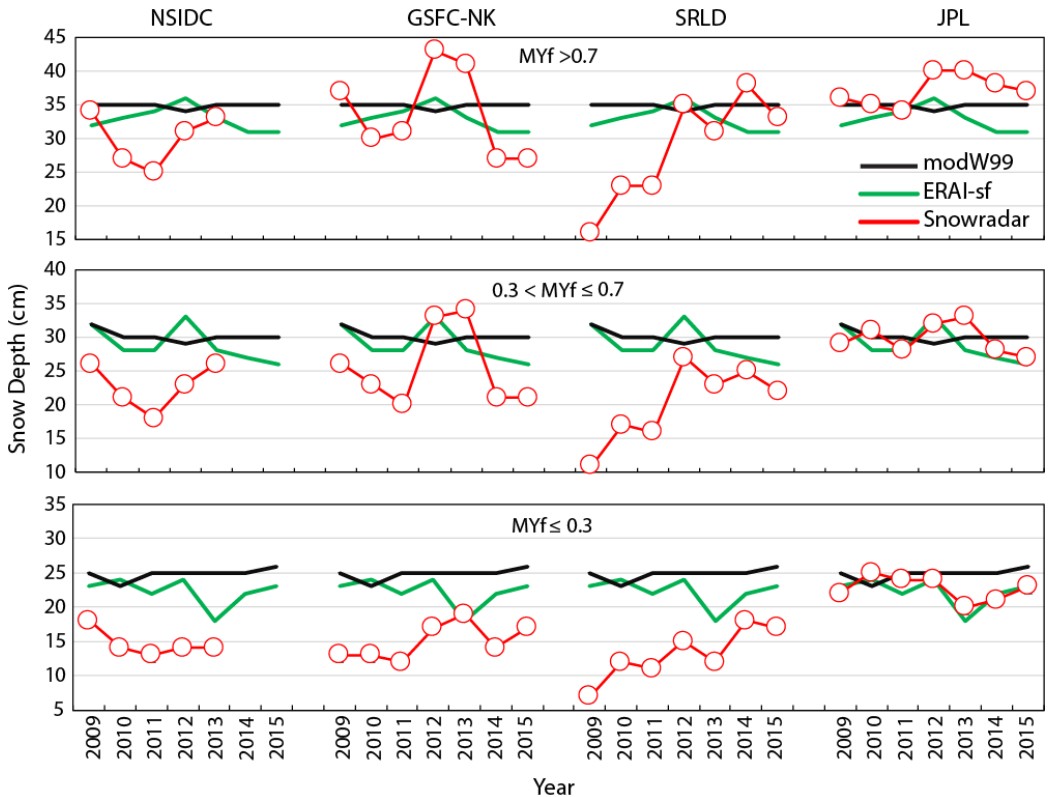

Figure 11. Comparison of snow depth retrievals from four algorithms (NSIDC, GSFC-NK, SRLD, JPL) with ERAI-sf and modW99 estimates in three multiyear/seasonal ice regimes (MYf ≥0.7; 0.3 < MYf < 0.7; MYf ≤ 0.3).





**Figure 12. Comparison of snow depth retrievals with ERAI-sf and modW99 estimates. (a) NSIDC. (b) GSFC-NK**





**Figure 13. Comparison of snow depth retrievals with ERAI-sf and modW99 estimates. (a) SRLD, (b) JPL.**