# Peer review of "Inter-comparison of snow depth retrievals over Arctic sea ice from radar data acquired by Operation IceBridge"

_The Cryosphere, 2017_

## Referee Comment (RC1) · Anonymous Referee #1 · 24 Jul 2017

General comments:

In the study "Inter-comparison of snow depth retrievals over Arctic sea ice from radar data acquired by Operation IceBridge", snow depth retrievals from five different algorithms are compared with each other, with in-situ snow depth measurements from two field surveys, a modified Warren climatology, and snow depths derived from ERA-Interim products. Most of the retrievals seem to reproduce the expected spatial snow depth patterns, certainly the gradient between first-year and multiyear ice. Nevertheless, differences between the products exist and biases can be significant. The authors state that the aim of this study is not to select the best algorithm, but to give an

informative overview of existing retrieval algorithms, also serving to develop the next-generation retrieval algorithm.

The manuscript fills an important gap in providing a comparative overview of the different Operation IceBridge snow depth retrievals that have been presented over the past years. The scientific analysis seems to be rigorous and the manuscript is well structured. Therefore, I do not really have any major objections. But I think the paper can be improved in terms of figure quality, and also some clarifications and more details are needed in some parts of the manuscript (see specific comments). For example, it is not always clear over which period the different retrievals exist. Sometimes, one retrieval is missing in the comparisons and figures. I guess this is because an algorithm was not applied during the time when the in-situ measurements took place? I would suggest to include an additional figure or table, showing the intervals of the 5 retrievals and the overlap with the in-situ measurements. Moreover, some figures lack a complete description, which sometimes makes it time-consuming to understand their content (see specific comments). I also wonder if the authors can provide some concluding remarks regarding the characteristics of the individual products in the conclusion section. To be more specific, GSFC-NK seems to reject much more echoes than other algorithms. Moreover some retrievals rather seem to over- or underestimate snow depths depending on how the interfaces are picked. Although I acknowledge that the aim of the paper is not to select the best algorithm, I think a statement about the characteristics of the different retrievals should be added in the conclusion.

Specific Comments:

P5L7: How does the automated snow depth probe work? Is there any reference to the instrument? I also couldn't find much information in Strum et al. (2006).

P7: What about the method and analysis in Holt et al. (2015)? Why is it not considered here? At least, I believe, results should be discussed and mentioned briefly in the introduction and/or discussion section.

P10L27: Why were the wavelet retrievals not available for the assessment? This question/comment aims to the one in the general comments. I think it would be useful to clarify the availability of the different retrievals, may be by including another figure or table.

P15L7: "work was not to the select…": delete "the"

P17L28: "Large interannual variability of retrievals from a given algorithm suggests issues in algorithmic robustness in adapting to changes in radar data quality." Could these signals not be (partly) real? Why should the interannual variability suggested by W99 be the reference?

Figure 2: a) Why is NSIDC missing here?

Figure 4: It would be helpful to add a legend in the figure explaining red and black dots rather than only mentioning it in the caption. And, there is a typo in the last sentence of the caption: "…are showN…".

Figure 5: I guess the error bars represent the standard deviation at certain cm bins? It is not really stated, neither in the caption. This should be added.

Figure 6: Why are there gaps in the histogram in d)?

Figure 9: a) Some points are beyond the axis, which looks a bit odd. Moreover, the number in the first column in the insets is not explained. I guess it is the number of measurements?

Figure 11: Why is the wavelet retrieval missing here (counts also for Figures 7, 8, 9 and 12)? b) I do not really understand the maps. Are the shown tracks the outbound and return tracks, but shifted to make differences visible? I would suggest to show just one map with the location of the tracks without color scale, and then ad xy along track plots with snow depths along outbound and return tracks.

---

## Referee Comment (RC2) · Anonymous Referee #2 · 27 Jul 2017

The paper provides an inter-comparison of five different NASA OIB snow depth products that have been developed in recent years. The OIB products are compared against in-situ data from two field campaigns and snow depth estimates derived from the ERA-Interim reanalysis and a modified snow climatology.

This work addresses an important issue for the Arctic sea ice community - what is the snow depth over Arctic sea ice and which (if any!) OIB derived snow depth product should we be using? The effort in bringing together these various snow groups and datasets is laudable. I believe the study should be published - we need to see these differences and have a baseline for snow inter-comparison discussions - but I believe

the paper has some shortcomings that need to be addressed first.

Main comments:

1. I think you need to discuss more the basin-scale differences between the products. I also think this should be moved further up the paper, as it is what motivates the whole exercise in my opinion, especially as you don't go into huge detail regarding the algorithm differences and echogram interface detection. It is also what most of the readers will be interested in seeing.

For example, I think the huge differences in the means across the products for the three ice classes should be its own figure near the start (although I also have issues with the use of age classes, see later main comment, so think this should be by region instead). There are differences of ∼100% between some products in the earlier OIB years, while some products show strong regional trends and others don't. It's pretty crazy to me that this is not discussed more as a motivating factor to look more closely at the algorithms, and I feel these differences have been somewhat hidden later in the paper. The more detailed in-situ comparisons could then follow on to help understand why this is.

2. Any comments on how 'tuned' these data have been, especially to the other snow depth data included in this paper? I believe I'm right in thinking the different groups have all had access to these in-situ data and ERA-I snow depth fields (especially as the lead author has previously produced the ERA-I derived snow depth maps used in the inter-comparison) so is that one reason why some fits are better than others? I understand that tuning happens and is often needed, but I think we need to understand this more to really understand if the differences are due to the choice of algorithm or other factors. Also, I think comments should be made if the individual algorithms were also compared against any other in-situ datasets in their respective papers and how good those fits were. The fact all authors are involved should make this easier.

3. As all the algorithm developers were part of the paper, I'm surprised a bigger comment was not made of what actually will happen next. Will one/multiple algorithms be scrapped or combined? Are there pros/cons of certain algorithms that will be adopted/used by the Operation IceBridge sea ice group? You do state in the paper that: "The aim of this paper is to examine these algorithms and to use the assessment results to inform the development of the next generation algorithm", but the path forward is unclear to me and I really hope we don't continue with multiple algorithms floating around that different groups/papers use for different reasons.

4. I'm not a huge fan of using the ice type mask to delineate the results. The comment on Page 14: " MYI that advected into this region, which was used in the construction of the modW99 fields but is absent in the ERAI-sf fields (because the MYI is not used in the estimates). " implies that the presence of MYI doesn't mean much for snow depth. I believe other cited studies (from co-authors) have come to similar conclusions (e.g. recent King and Webster papers). The modified Warren climatology seems just plain wrong in my opinion so I would be tempted to drop that entirely unless you want to make the point that some groups are using this now and we need to explore its potential biases.

5. Why were only these specific field campaigns chosen?

6. Why ERA-Interim for derived snowfall?

7. Why were the Wavelet retrievals not available? The Newman et al., (2014) paper shows that data were produced in 2012..? This seems odd.

Specific Comments:

P2, L14 - maybe 'needs to be inferred by other methods' instead of left to be measured or modeled

P2, L14 - I think (if I've interpreted this right) that you should say why snow density matters before saying we need routine measurements of it.

P2, L16 - you say hence, but then start by discussing forecasting, which seems odd.

P2, L20 - I think more should be made of the fact that people still use this climatology, despite it being many decades old!

P2, L23-25 - 'of about several centimeters' and 'broadly consistent' seems pretty loose. Drop or further clarify.

P2, L26 - mention that this is predominantly the western Arctic, (except for 2017 which isn't included in this study).

P2, L29 - add something like 'from the OIB snow radar..'

P5, L30 - this sentence is poorly worded.

P6, L4 - why and how did it vary with ice topography? Just because it was older

do we think this is an exhaustive list of retrieval algorithms?

P8, L12 - unsure of the comment " The initial application to existing OIB snow radar data from various campaigns (2009 - 2012) and the need for it to be applicable to future campaigns, required a process that would adapt to the data and not be dependent on fixed thresholds in the radar return signal ". Why can't you apply thresholds and update these each year when you process the data?

P9, L6 - how is it robust? It seems we are testing that in this paper, no? What do you mean by this?

P9 - Does the removal of deformed ice from the Wavelet algorithm introduce a bias compared to other algorithms?

P9, L20 - is this the only thing that has been removed from the algorithm?

P10, L11 - this should be Figure 3.

P10, L15 - I'm a bit confused by this. the resolution of each radar footprint is around 5-10 m, right? So how does a 20 m radius circle correspond to 9 radar spots again?

P10, L18 - you mean the mean AND standard deviation, right? In which case I don't get

how using an averaging window changes the mean value. An average of an average should produce the same average..? It should obviously change the shape of the distribution though (reducing the tail).

P11, L25 - can you list them here? It looks like the JPL and Wavelet correlations decrease. Any comment on how this compares with the htopo parameter? i.e. do you expect surface roughness to co-vary with the snow depth variability?

P12, L14 why only four of five algorithms? Pretty interested to see the NSIDC differences, especially as this is probably the most commonly used..?

P12, L27 - this seems the fundamental tenet of the whole paper, no?!

P14, L20 onwards - this should go at the start of the section in my mind as it's a pretty key point.

Interpreting Table 2 and 3 was pretty painstaking at first. Can you make it more obvious that the diagonal elements are taken from Table 2 and maybe draw a box around these?

Figure 9 - Confused by the numbers in Figure 9a. There is a lot of information being crammed in and I struggled to understand what it all means.

- Why is this saturated at 15 cm?

- Why are the NSIDC panels missing the repeat tracks and distributions?

Multiple figures - The Jet color scale introduces false boundaries, isn't good for people with colorblindness, and should thus not be used in my opinion! Very bad for comparing geospatial data by eye.

Figure 10 - this seems pretty pointless so I would be inclined to drop it.

Figure 12 and 13 should be split up and made more readable.

---

## Author Comment (AC1) · 31 Aug 2017

The comment was uploaded in the form of a supplement:
https://www.the-cryosphere-discuss.net/tc-2017-103/tc-2017-103-AC1-supplement.pdf

---

## Author Comment (AC2) · 31 Aug 2017

**Anonymous Referee #1 ((Referee comments are in italics)**

*General comments:*
*In the study "Inter-comparison of snow depth retrievals over Arctic sea ice from radar data acquired by Operation IceBridge", snow depth retrievals from five different algorithms are compared with each other, with in-situ snow depth measurements from two field surveys, a modified Warren climatology, and snow depths derived from ERA-Interim products. Most of the retrievals seem to reproduce the expected spatial snow depth patterns, certainly the gradient between first-year and multiyear ice. Nevertheless, differences between the products exist and biases can be significant. The authors state that the aim of this study is not to select the best algorithm, but to give an informative overview of existing retrieval algorithms, also serving to develop the next-generation retrieval algorithm.*
*The manuscript fills an important gap in providing a comparative overview of the different Operation IceBridge snow depth retrievals that have been presented over the past years. The scientific analysis seems to be rigorous and the manuscript is well structured. Therefore, I do not really have any major objections. But I think the paper can be improved in terms of figure quality, and also some clarifications and more details are needed in some parts of the manuscript (see specific comments). For example, it is not always clear over which period the different retrievals exist. Sometimes, one retrieval is missing in the comparisons and figures. I guess this is because an algorithm was not applied during the time when the in-situ measurements took place? I would suggest to include an additional figure or table, showing the intervals of the 5 retrievals and the overlap with the in-situ measurements.*
*Moreover, some figures lack a complete description, which sometimes makes it time-consuming to understand their content (see specific comments). I also wonder if the authors can provide some concluding remarks regarding the characteristics of the individual products in the conclusion section. To be more specific, GSFC-NK seems to reject much more echoes than other algorithms. Moreover some retrievals rather seem to over- or underestimate snow depths depending on how the interfaces are picked. Although I acknowledge that the aim of the paper is not to select the best algorithm, I think a statement about the characteristics of the different retrievals should be added in the conclusion.*

Thank you for your suggestions.

In the revised manuscript, we now include a table that shows the availability of retrievals for each algorithm at the time the inter-comparisons were carried out. We also note that there is only one standard product (archived at NSIDC) provided by the OIB project, and other snow depth estimates used here (Wavelet, SRLD, JPL) were contributed by scientists interested in the retrieval process. The contributed retrievals and the results here will be used to guide the next step in the development of a community retrieval approach. At that time, there will be a detailed description of the new approach used in the processing of the snow radar data.

*Specific Comments:*
*P5L7: How does the automated snow depth probe work? Is there any reference to the instrument? I also couldn't find much information in Strum et al. (2006).*
A magnaprobe is an automated snow depth probe, approximately ~1.5 m in length, connected to a GPS and data logger in a backpack. At the end of the steel probe is an adjustable white basket, which sits on

top of the snow surface as the probe enters the snow pack. The depth is recorded as the distance between the tip of the snow probe and the white basket at the surface. GPS coordinates of each snow depth measurement are simultaneously recorded in the data logger. We now provide a citation to a document that describes the patented technology (Sturm and Holmgren, 1999).

*P7: What about the method and analysis in Holt et al. (2015)? Why is it not considered here? At least, I believe, results should be discussed and mentioned briefly in the introduction and/or discussion section.*
The authors of that algorithm elected not to participate in the inter-comparison project. We have added a citation to the work of Holt et al. in the text.

*P10L27: Why were the wavelet retrievals not available for the assessment? This question/comment aims to the one in the general comments. I think it would be useful to clarify the availability of the different retrievals, may be by including another figure or table.*
We added a table that shows the availability of retrievals from each algorithm for the intercomparisons period.

*P15L7: "work was not to the select...": delete "the"*
Done.

*P17L28: "Large interannual variability of retrievals from a given algorithm suggests issues in algorithmic robustness in adapting to changes in radar data quality." Could these signals not be (partly) real? Why should the interannual variability suggested by W99 be the reference?*
Yes, not all of the retrievals are in error - it is only that the basin-scale averages may be biased by those retrievals affected by system artifacts and changes in radar data quality.
We expect extreme deviations from modW99 and ERAI-sf to be suspect given:
1. the expected interannual variability of climatological snow depth of ~6 cm.
2. the relative agreement between modW99 and ERAI-sf over the period.

*Figure 2: a) Why is NSIDC missing here?*
The reason is that the standard NSIDC products are not sampled at the correct rate for these plots.

*Figure 4: It would be helpful to add a legend in the figure explaining red and black dots rather than only mentioning it in the caption. And, there is a typo in the last sentence of the caption: "...are showN...".*
Legend added to the figure and typo corrected.

*Figure 5: I guess the error bars represent the standard deviation at certain cm bins? It is not really stated, neither in the caption. This should be added.*
Added.

*Figure 6: Why are there gaps in the histogram in d)?*
The gaps are likely introduced by discretization of the snow depth estimates.

*Figure 9: a) Some points are beyond the axis, which looks a bit odd. Moreover, the number in the first column in the insets is not explained. I guess it is the number of measurements?*
The quantities in the insets are now explained (they are the mean and standard deviation of the differences, the correlation of snow depth between the two tracks, and the number of samples).

*Figure 11: Why is the wavelet retrieval missing here (counts also for Figures 7, 8, 9 and 12)?*

The wavelet retrievals were not available at the time of this work.

*b) I do not really understand the maps. Are the shown tracks the outbound and return tracks, but shifted to make differences visible? I would suggest to show just one map with the location of the tracks without color scale, and then ad xy along track plots with snow depths along outbound and return tracks.*

The tracks are not shifted in these figures. The only repeated outbound and return tracks (i.e., near coincident tracks) are shown in Figure 9. This is clarified in the caption of Figure 7.

---

## Author Comment (AC3) · 31 Aug 2017

**Anonymous Referee #2 (Referee comments are in italics)**

*The paper provides an inter-comparison of five different NASA OIB snow depth products that have been developed in recent years. The OIB products are compared against in-situ data from two field campaigns and snow depth estimates derived from the ERA-Interim reanalysis and a modified snow climatology.*

*This work addresses an important issue for the Arctic sea ice community what is the snow depth over Arctic sea ice and which (if any!) OIB derived snow depth product should we be using? The effort in bringing together these various snow groups and datasets is laudable. I believe the study should be published we need to see these differences and have a baseline for snow inter-comparison discussions - but I believe the paper has some shortcomings that need to be addressed first.*

We thank the reviewer for the detailed reading of the manuscript.

We would like to note that there is only one standard product (archived at NSIDC) provided by the OIB project, and other snow depth estimates used here (Wavelet, SRLD, JPL) were contributed by scientists interested in the retrieval process. We now include a table (Table 2) that shows the availability of retrievals for each algorithm at the time of the inter-comparisons were carried out.

*Main comments:*

*1. I think you need to discuss more the basin-scale differences between the products. I also think this should be moved further up the paper, as it is what motivates the whole exercise in my opinion, especially as you don't go into huge detail regarding the algorithm differences and echogram interface detection. It is also what most of the readers will be interested in seeing.*

*For example, I think the huge differences in the means across the products for the three ice classes should be its own figure near the start (although I also have issues with the use of age classes, see later main comment, so think this should be by region instead). There are differences of ~100% between some products in the earlier OIB years, while some products show strong regional trends and others don't. It's pretty crazy to me that this is not discussed more as a motivating factor to look more closely at the algorithms, and I feel these differences have been somewhat hidden later in the paper. The more detailed in-situ comparisons could then follow on to help understand why this is.*

We appreciate the broader geophysical perspective although this was not the path taken by the inter-comparison project. Initially, the project was more interested in the quality of different retrieval approaches when assessed with *in situ* snow depth because that allowed for more quantitative evaluations of the procedures at the highest spatial resolution available (i.e., small scale variability). The merits of the basin-scale comparisons were recognized only after the results of the spatial and inter-annual differences were produced. We noted in the text that the robustness and adaptation of the retrieval procedures to changes in radar data quality over the IceBridge Mission are important considerations, in addition to the footprint-scale comparisons, in producing a long-term record.

We have added to the text to provide a better description of the evolution of the project (i.e., from the small scale to the large, rather than the motivation suggested above) but we prefer to preserve the order of the discussion in the manuscript.

*2. Any comments on how 'tuned' these data have been, especially to the other snow depth data included in this paper? I believe I'm right in thinking the different groups have all had access to these in-situ data and ERA-I snow depth fields (especially as the lead author has previously produced the ERA-I derived snow depth maps used in the inter-comparison) so is that one reason why some fits are better than others? I understand that tuning happens and is often needed, but I think we need to understand this more to really understand if the differences are due to the choice of algorithm or other factors. Also, I think comments should be made if the individual algorithms were also compared against any other in-situ datasets in their respective papers and how good those fits were. The fact all authors are involved should make this easier.*

The snow depth retrievals were contributed by different algorithm-developers. Thus, there was no control on the amount of 'tuning'. It was entirely up to the developers of the snow depth data sets. The level of maturity of the algorithms is different and depends on the amount of resources available to the developers. The aim of the work was not to the select the best algorithm, but rather to provide results that would serve to inform the development of the next-generation retrieval algorithm.

*3. As all the algorithm developers were part of the paper, I'm surprised a bigger comment was not made of what actually will happen next. Will one/multiple algorithms be scrapped or combined? Are there pros/cons of certain algorithms that will be adopted/used by the Operation IceBridge sea ice group? You do state in the paper that: "The aim of this paper is to examine these algorithms and to use the assessment results to inform the development of the next generation algorithm", but the path forward is unclear to me and I really hope we don't continue with multiple algorithms floating around that different groups/papers use for different reasons.*

The next step is to develop an improved algorithm, for producing an OIB product, by integrating the experience gained from this work.

*4. I'm not a huge fan of using the ice type mask to delineate the results. The comment on Page 14: " MYI that advected into this region, which was used in the construction of the modW99 fields but is absent in the ERAI-sf fields (because the MYI is not used in the estimates). " implies that the presence of MYI doesn't mean much for snow depth. I believe other cited studies (from co-authors) have come to similar conclusions (e.g. recent King and Webster papers). The modified Warren climatology seems just plain wrong in my opinion so I would be tempted to drop that entirely unless you want to make the point that some groups are using this now and we need to explore its potential biases.*

There is merit in using multiyear ice (i.e., ice that survived the summer in this case) as a gross indicator of the chronological age of the ice measured from the 'beginning' of the growth season; one expects more snow to accumulate on older ice (on average) and hence a thicker snow cover on this ice type. We have clarified this in the text as this usage is somewhat different than the way we typically think about ice age (i.e., in terms of years rather than from the beginning of the season).

Regarding modified Warren climatology, the modification adapts to the thinner snow cover over seasonal ice. Some form of modified climatology is used by most of the sea ice thickness algorithm at different institutions.

*5. Why were only these specific field campaigns chosen?*

These were the only field campaigns over fast ice, where we did not have to deal with spatial registration issues related to sea ice motion.

*6. Why ERA-Interim for derived snowfall?*

We considered MERRA2 as well but the snowfall from MERRA2 is known be biased (higher by ~30-40%) compared to climatology and ERA-Interim.

*7. Why were the Wavelet retrievals not available? The Newman et al., (2014) paper shows that data were produced in 2012..? This seems odd.*

We used only those data sets that were available and provided by the algorithm developers at the time of this inter-comparison project. In the case of the Wavelet retrievals, the algorithm developers provided only retrievals from the flight over the Eureka field campaign.

*Specific Comments:*

*P2, L14 - maybe 'needs to be inferred by other methods' instead of left to be measured or modeled*

We prefer the way it is currently phased because it suggests the two alternatives for obtaining snow depth.

*P2, L14 - I think (if I've interpreted this right) that you should say why snow density matters before saying we need routine measurements of it.*

Added a note about snow density.

*P2, L16 - you say hence, but then start by discussing forecasting, which seems odd.*

Re-ordered.

*P2, L20 - I think more should be made of the fact that people still use this climatology, despite it being many decades old!*

Added: "…The *W99* climatology is still widely used in ice thickness retrievals.*"*

*P2, L23-25 - 'of about several centimeters' and 'broadly consistent' seems pretty loose. Drop or further clarify.*

The discussion is a broad summary of the results from the list of papers provided at the end of this sentence.

*P2, L26 - mention that this is predominantly the western Arctic, (except for 2017 which isn't included in this study).*

Revised to read: "…repeat surveys of the early spring snow and ice conditions in different parts of the western Arctic.

*P2, L29 - add something like 'from the OIB snow radar..'*

Revised to read: "…These snow depth datasets from the OIB snow radar…"

*P5, L30 - this sentence is poorly worded.*

Reworded.

*P6, L4 - why and how did it vary with ice topography? Just because it was older do we think this is an exhaustive list of retrieval algorithms?*

Modified to read: "Transect variations in density were conservative, with a mean of 306 kg m$^{-3}$ and standard deviation of 50 kg m$^{-3}$, comparable to the assumed climatological mean of ~320 kg m$^{-3}$ near the end of the winter."

*P8, L12 - unsure of the comment " The initial application to existing OIB snow radar data from various campaigns (2009 - 2012) and the need for it to be applicable to future campaigns, required a process that would adapt to the data and not be dependent on fixed thresholds in the radar return signal ". Why can't you apply thresholds and update these each year when you process the data?*

The comment emphasizes the need for a procedure that is independent of fixed thresholds is desirable (and arguably necessary) for several reasons. For the SRLD algorithm, there is no need to change threshold values depending on which data set is being analyzed, whether it is for the latest campaign, or the multiple data sets within an archive of previous campaigns (e.g. 2009-2015). More significantly, there is no need to determine those threshold values, which would need to be done empirically by analyzing each data set beforehand. Additionally, there is no need to determine when to determine new threshold values. That is, a change in calibration could conceivably occur within a campaign for example.

*P9, L6 - how is it robust? It seems we are testing that in this paper, no? What do you mean by this? P9 - Does the removal of deformed ice from the Wavelet algorithm introduce a bias compared to other algorithms?*

For a more detailed description of the Wavelet algorithm, the reviewer is referred to Newman et al. (2014).

*P9, L20 - is this the only thing that has been removed from the algorithm?*

Yes.

*P10, L11 - this should be Figure 3.*

Corrected.

*P10, L15 - I'm a bit confused by this. the resolution of each radar footprint is around 5-10 m, right? So how does a 20 m radius circle correspond to 9 radar spots again?*

There are approximately nine radar footprints (sampled at 5-meter intervals) in a 40-m along track segment. The aim is to reduce the geophysical variability as well as the sensitivity to accommodate for uncertainties in the spatial overlap between the snow-radar footprint and the point samples from the field measurements. This is discussed in the text.

*P10, L18 - you mean the mean AND standard deviation, right? In which case I don't get how using an averaging window changes the mean value. An average of an average should produce the same average..? It should obviously change the shape of the distribution though (reducing the tail).*

It is the mean standard deviation (i.e., the mean of the distribution of standard deviations), not the mean AND standard deviation.

Added to clarify:"… (i.e., mean of the distribution of $\sigma_f$).."

*P11, L25 - can you list them here? It looks like the JPL and Wavelet correlations decrease. Any comment on how this compares with the htopo parameter? i.e. do you expect surface roughness to co-vary with the snow depth variability?*

1. The changes in the correlation values are now listed in the text.

2. These comparisons do not make use of specific metrics provided by each algorithm.

*P12, L14 why only four of five algorithms? Pretty interested to see the NSIDC differences, especially as this is probably the most commonly used..?*

The NSIDC products are averaged spatially and do not provide snow depth estimates for each echogram, which was needed for the plots shown in Figure 3b.

*P12, L27 - this seems the fundamental tenet of the whole paper, no?!*

Yes!

*P14, L20 onwards - this should go at the start of the section in my mind as it's a pretty key point.*

We prefer order of the current list as most of points are important things we have learned in this process.

*Interpreting Table 2 and 3 was pretty painstaking at first. Can you make it more obvious that the diagonal elements are taken from Table 2 and maybe draw a box around these?*

Yes – we have clarified this in the caption. Also, quantities from Table 2 are now italicized.

*Figure 9 - Confused by the numbers in Figure 9a. There is a lot of information being crammed in and I struggled to understand what it all means.*

*- Why is this saturated at 15 cm?*

  15 cm was selected because the threshold of detectability of the a-s interface is ~10 cm.

*- Why are the NSIDC panels missing the repeat tracks and distributions?*

  The repeat tracks are not processed by the NSIDC algorithm.

*Multiple figures - The Jet color scale introduces false boundaries, isn't good for people with colorblindness, and should thus not be used in my opinion! Very bad for comparing geospatial data by eye.*

It is somewhat difficult to control the quality of the figures in the pdf files generated by the publisher for review purposes. We have enlarged the tracks in Figure 9 so that they are easier to see. The quality in the final publication should be higher.

*Figure 10 - this seems pretty pointless so I would be inclined to drop it. Figure 12 and 13 should be split up and made more readable.*

This figure shows the differences between ERAI-sf and modW99. We believe that it is a useful illustration of the spatial differences of the two fields. Figures 12 and 13 are now in four separate pages.

---

## Referee Report (RR1)

Second review of "**Inter-comparison of snow depth retrievals over Arctic sea ice from
radar data acquired by Operation IceBridge "** by Kwok et al

I appreciate the efforts of the author in responding to my earlier comments. I feel some of the replies are unsatisfactory and thus request some more clarification and also clear additions to the manuscript following these responses, before I believe this to be ready for publication.

My new reviewer comments are in red, in response to the response document copied below. I only included comments that required a second response by me.

*Main comments:*
*1. I think you need to discuss more the basin-scale differences between the products. I also think this should be moved further up the paper, as it is what motivates the whole exercise in my opinion, especially as you don't go into huge detail regarding the algorithm differences and echogram interface detection. It is also what most of the readers will be interested in seeing.*
*For example, I think the huge differences in the means across the products for the three ice classes should be its own figure near the start (although I also have issues with the use of age classes, see later main comment, so think this should be by region instead). There are differences of ~100% between some products in the earlier OIB years, while some products show strong regional trends and others don't. It's pretty crazy to me that this is not discussed more as a motivating factor to look more closely at the algorithms, and I feel these differences have been somewhat hidden later in the paper. The more detailed in-situ comparisons could then follow on to help understand why this is.*
We appreciate the broader geophysical perspective although this was not the path taken by the inter-comparison project. Initially, the project was more interested in the quality of different retrieval approaches when assessed with *in situ* snow depth because that allowed for more quantitative evaluations of the procedures at the highest spatial resolution available (i.e., small scale variability). The merits of the basin-scale comparisons were recognized only after the results of the spatial and inter-annual differences were produced. We noted in the text that the robustness and adaptation of the retrieval procedures to changes in radar data quality over the IceBridge Mission are important considerations, in addition to the footprint-scale comparisons, in producing a long-term record. We have added to the text to provide a better description of the evolution of the project (i.e., from the small scale to the large, rather than the motivation suggested above) but we prefer to preserve the order of the discussion in the manuscript.

Can you indicate how you have done this please? I.e. what you added and where.

*2. Any comments on how 'tuned' these data have been, especially to the other snow depth data included in this paper? I believe I'm right in thinking the different groups have all had access to these in-situ data and ERA-I snow depth fields (especially as the lead author has previously produced the ERA-I derived snow depth maps used in the inter-comparison) so is that one reason why some fits are better than others? I understand that*

*tuning happens and is often needed, but I think we need to understand this more to really understand if the differences are due to the choice of algorithm or other factors. Also, I think comments should be made if the individual algorithms were also compared against any other in-situ datasets in their respective papers and how good those fits were. The fact all authors are involved should make this easier.*

The snow depth retrievals were contributed by different algorithm-developers. Thus, there was no control on the amount of 'tuning'. It was entirely up to the developers of the snow depth data sets. The level of maturity of the algorithms is different and depends on the amount of resources available to the developers. The aim of the work was not to the select the best algorithm, but rather to provide results that would serve to inform the development of the next-generation retrieval algorithm.

OK, but as all the developers are part of this paper, it should be possible to provide some factual statements regarding this and to include a discussion in the manuscript regarding the issue. This is a crucial point in terms of reproducibility and understanding how/why better correlations with in-situ/reanalysis data were found, which I feel is still inadequately treated.

*3. As all the algorithm developers were part of the paper, I'm surprised a bigger comment was not made of what actually will happen next. Will one/multiple algorithms be scrapped or combined? Are there pros/cons of certain algorithms that will be adopted/used by the Operation IceBridge sea ice group? You do state in the paper that: "The aim of this paper is to examine these algorithms and to use the assessment results to inform the development of the next generation algorithm", but the path forward is unclear to me and I really hope we don't continue with multiple algorithms floating around that different groups/papers use for different reasons.*

The next step is to develop an improved algorithm, for producing an OIB product, by integrating the experience gained from this work.

It is still not clear to me that this paper has made a significant step towards this other than highlighting the (albeit important) differences between the current algorithms, but it doesn't seem like a more concrete statement will be forthcoming.

*5. Why were only these specific field campaigns chosen?*

These were the only field campaigns over fast ice, where we did not have to deal with spatial registration issues related to sea ice motion.

Can you include this comment in the manuscript?

*6. Why ERA-Interim for derived snowfall?*

We considered MERRA2 as well but the snowfall from MERRA2 is known be biased (higher by ~30-40%) compared to climatology and ERA-Interim.

OK but there are other reanalyses available that might not have such a bias. Obviously some also don't provide snowfall which may be an issue? At least a comment on this would be useful (I don't expect you to add in any analysis on this at this stage).

*7. Why were the Wavelet retrievals not available? The Newman et al., (2014) paper shows that data were produced in 2012..? This seems odd.*
We used only those data sets that were available and provided by the algorithm developers at the time of this inter-comparison project. In the case of the Wavelet retrievals, the algorithm developers provided only retrievals from the flight over the Eureka field campaign.

This seems very unusual, although I don't expect you to change this at this stage.

*Figure 9 - Confused by the numbers in Figure 9a. There is a lot of information being crammed in and I struggled to understand what it all means.*
*- Why is this saturated at 15 cm?* 15 cm was selected because the threshold of detectability of the a-s interface is ~10 cm.

So this should be 10 cm then.

*Multiple figures - The Jet color scale introduces false boundaries, isn't good for people with colorblindness, and should thus not be used in my opinion! Very bad for comparing geospatial data by eye.*
It is somewhat difficult to control the quality of the figures in the pdf files generated by the publisher for review purposes. We have enlarged the tracks in Figure 9 so that they are easier to see. The quality in the final publication should be higher.

This is nothing to do with how the pdf is generated but the use of a bad color scale that makes it harder to interpret the figure data values, no matter how it's generated into the pdf. See e.g. here https://www.climate-lab-book.ac.uk/2014/end-of-the-rainbow/ for a discussion.

---

## Author Response (AR2)

**Anonymous Referee #2** (*Authors' responses are in green italics*)

Second review of "**Inter-comparison of snow depth retrievals over Arctic sea ice from radar data acquired by Operation IceBridge** " by Kwok et al

I appreciate the efforts of the author in responding to my earlier comments. I feel some of the replies are unsatisfactory and thus request some more clarification and also clear additions to the manuscript following these responses, before I believe this to be ready for publication.

My new reviewer comments are in red, in response to the response document copied below. I only included comments that required a second response by me.

*Main comments:*
*1. I think you need to discuss more the basin-scale differences between the products. I also think this should be moved further up the paper, as it is what motivates the whole exercise in my opinion, especially as you don't go into huge detail regarding the algorithm differences and echogram interface detection. It is also what most of the readers will be interested in seeing.*
*For example, I think the huge differences in the means across the products for the three ice classes should be its own figure near the start (although I also have issues with the use of age classes, see later main comment, so think this should be by region instead). There are differences of ~100% between some products in the earlier OIB years, while some products show strong regional trends and others don't. It's pretty crazy to me that this is not discussed more as a motivating factor to look more closely at the algorithms, and I feel these differences have been somewhat hidden later in the paper. The more detailed in-situ comparisons could then follow on to help understand why this is.*
We appreciate the broader geophysical perspective although this was not the path taken by the inter-comparison project. Initially, the project was more interested in the quality of different retrieval approaches when assessed with *in situ* snow depth because that allowed for more quantitative evaluations of the procedures at the highest spatial resolution available (i.e., small scale variability). The merits of the basin-scale comparisons were recognized only after the results of the spatial and inter-annual differences were produced. We noted in the text that the robustness and adaptation of the retrieval procedures to changes in radar data quality over the IceBridge Mission are important considerations, in addition to the footprint-scale comparisons, in producing a long-term record. We have added to the text to provide a better description of the evolution of the project (i.e., from the small scale to the large, rather than the motivation suggested above but we prefer to preserve the order of the discussion in the manuscript.

Can you indicate how you have done this please? I.e. what you added and where.

*In the last paragraph of the introduction in our revision, we provided a roadmap of the paper:*
*"…The comparisons with field measurements allow a detailed assessment of the retrievals locally, while the comparisons with climatology and analyzed snowfall provide a large-scale multi-year perspective of their year-to-year retrieval consistency and robustness to changes in radar parameters, and their relative agreements with basin-scale fields."*

*We do not feel that it adds to the manuscript to describe the detailed evolution of the project.*

*2. Any comments on how 'tuned' these data have been, especially to the other snow depth data included in this paper? I believe I'm right in thinking the different groups have all had access to these in-situ data and ERA-I snow depth fields (especially as the lead author has previously produced the ERA-I derived snow depth maps used in the inter- comparison) so is that one reason why some fits are better than others? I understand that tuning happens and is often needed, but I think we need to understand this more to really understand if the differences are due to the choice of algorithm or other factors. Also, I think comments should be made if the individual algorithms were also compared against any other in-situ datasets in their respective papers and how good those fits were. The fact all authors are involved should make this easier.*
The snow depth retrievals were contributed by different algorithm-developers. Thus, there was no control on the amount of 'tuning'. It was entirely up to the developers of the snow depth data sets. The level of maturity of the algorithms is different and depends on the amount of resources available to the developers. The aim of the work was not to the select the best algorithm, but rather to provide results that would serve to inform the development of the next-generation retrieval algorithm.

OK, but as all the developers are part of this paper, it should be possible to provide some factual statements regarding this and to include a discussion in the manuscript regarding the issue. This is a crucial point in terms of reproducibility and understanding how/why better correlations with in-situ/reanalysis data were found, which I feel is still inadequately treated.

*We do not have a fixed set of parameters for each algorithm that one could control because the approaches are very different, and thus it is difficult to comment on the 'tuning' aspects that were undertaken by individual developers. The objective of this work was a broad assessment based on the differences in the localization of the air-snow and snow-ice interfaces without considering the details of the approaches (illustrated in Figure 6). We reiterate that there was NO control over parameters chosen and it was up to the developers on what was delivered for assessment. Yes, the conclusions are rather broad and it remains one of the challenges for the group to harmonize the different approaches. (see also response to next comment)*

*3. As all the algorithm developers were part of the paper, I'm surprised a bigger comment was not made of what actually will happen next. Will one/multiple algorithms be scrapped or combined? Are there pros/cons of certain algorithms that will be adopted/used by the Operation IceBridge sea ice group? You do state in the paper that: "The aim of this paper is to examine these algorithms and to use the assessment results to inform the development of the next generation algorithm", but the path forward is unclear to me and I really hope we don't continue with multiple algorithms floating around that different groups/papers use for different reasons.*
The next step is to develop an improved algorithm, for producing an OIB product, by integrating the experience gained from this work.

It is still not clear to me that this paper has made a significant step towards this other than highlighting the (albeit important) differences between the current algorithms, but it doesn't seem like a more concrete statement will be forthcoming.

*With a group of investigators, because of individual preferences in algorithm development, moving forward involves first building consensus and agreement prior to taking the next steps.*

*We believe that this inter-comparison paper served these goals. The next step is to incorporate the merits of each algorithm in future developments. As mentioned above, it remains one of the challenges for the group to harmonize the different approaches and to provide an improved data product.*

*In the conclusion, we have added:"... The next step is to harmonize the differences in the present approaches and to provide an improved snow depth product..."*

5. *Why were only these specific field campaigns chosen?*
These were the only field campaigns over fast ice, where we did not have to deal with spatial registration issues related to sea ice motion.

Can you include this comment in the manuscript?

*This statement was included in the first paragraph of Section 2.2 in our last revision: "...Coordinated surveys of field measurements and OIB overflights occurred in two years (2012 and 2014) under the auspices of two different programs. Both were located on landfast ice to minimize to the variability introduced by the spatial mismatch of the airborne and ground-based measurements due to ice drift ..."*

6. *Why ERA-Interim for derived snowfall?*
We considered MERRA2 as well but the snowfall from MERRA2 is known to be biased (higher by ~30-40%) compared to climatology and ERA-Interim.

OK but there are other reanalyses available that might not have such a bias. Obviously some also don't provide snowfall which may be an issue? At least a comment on this would be useful (I don't expect you to add in any analysis on this at this stage).

*It is not the intent of this manuscript to provide an exhaustive assessment of all available reanalyses, suffice to say that ERA-Interim is one of the more broadly used reanalysis in climate work and compares reasonably with the modified climatology. We prefer not to comment directly on MERRA2 or some other reanalysis because we are not aware of any published work that addresses this topic of snowfall and snow depth.*

*The first paragraph of Section 2.3 has been modified to read: "Averaged retrievals (25 km × 25 km) from the five algorithms are compared with both snow depths from snowfall in ERA-Interim products and a modification of the W99 climatology. We elected to use the ERA-Interim products here because it is one of the more broadly used reanalysis in climate work and compares reasonably with the modified climatology (see Section 4). Below, the procedures for constructing these daily fields of basin-wide snow depths are described."*

7. *Why were the Wavelet retrievals not available? The Newman et al., (2014) paper shows that data were produced in 2012..? This seems odd.*
We used only those data sets that were available and provided by the algorithm developers at the time of this inter-comparison project. In the case of the Wavelet retrievals, the algorithm developers provided only retrievals from the flight over the Eureka field campaign.

This seems very unusual, although I don't expect you to change this at this stage.

*We had a specific timeframe/schedule for completion of the project and had to set a fixed delivery date for the developers so that the project could proceed. The investigators who were involved in the development of the Wavelet retrievals were not able to provide their dataset during the agreed upon date. We chose to include what was available at the time and that is why we have only a subset of this data set to show. It is unfortunate but it is not fair for the entire group to include the Wavelet retrievals, which to this date have not been delivered.*

*Figure 9 - Confused by the numbers in Figure 9a. There is a lot of information being crammed in and I struggled to understand what it all means.*
*- Why is this saturated at 15 cm?* 15 cm was selected because the threshold of detectability of the a-s interface is ~10 cm.

So this should be 10 cm then.
*No, the 15-55 cm range was chosen for best presentation of the datasets in hand. And, it is not just the lower bound physical limit of the retrievals.*

*Multiple figures - The Jet color scale introduces false boundaries, isn't good for people with colorblindness, and should thus not be used in my opinion! Very bad for comparing geospatial data by eye.*
It is somewhat difficult to control the quality of the figures in the pdf files generated by the publisher for review purposes. We have enlarged the tracks in Figure 9 so that they are easier to see. The quality in the final publication should be higher.

This is nothing to do with how the pdf is generated but the use of a bad color scale that makes it harder to interpret the figure data values, no matter how it's generated into the pdf. See e.g. here https://www.climate-lab-book.ac.uk/2014/end-of-the-rainbow/ for a discussion.

*We take to heart the color-scale issue that the reviewer alluded to and will carefully consider our choice in future publications, but we elect to retain the colors used by our figures at this time.*

[revised manuscript text omitted]